

# Skilful Seasonal Streamflow Forecasting Using a Fully Coupled Global Climate Model

Gabriel Fernando Narváez-Campo[1] and Constantin Ardilouze[1]

[1]CNRM, Université de Toulouse, Météo-France, CNRS, Toulouse, France

**Correspondence:** gabriel.narvaez-campo@umr-cnrm.fr)

**Abstract.** The seasonal streamflow forecast (SSF) is a crucial decision-making, planning and management tool for disaster prevention, navigation, agriculture, and hydropower generation. This study demonstrates for the first time the capacity of a fully coupled operational global forecast system to directly provide skilful seasonal streamflow predictions through a physically consistent and convenient single-step workflow for forecast production. We assess the skill of the SSF derived from the operational Météo France forecast system SYS8, based on the *in-house* fully coupled atmosphere-ocean-land general circulation model of the sixth generation, CNRM-CM6-1. An advanced river routing model interacts with the land and atmosphere *via* surface/sub-surface runoff, aquifer exchange and open water evaporation to predict river streamflow. The actual skill is evaluated against streamflow observations, with the Ensemble Streamflow Prediction (ESP) approach used as a benchmark. Results show that the online coupled forecast system is overall more skilful than ESP in predicting streamflow for the summer and winter seasons. This improvement is particularly notable with enhanced land water storage initial conditions, especially in summer and in large basins where the low-flow response is influenced by soil water storage. Predicting climate anomalies is crucial in winter forecasting, and results consistently suggest that the atmospheric forecast of the fully coupled CNRM-CM6-1 model contributes to better seasonal streamflow forecasts than the climatology-based ESP benchmark. This study showcases the capacity of an operational seasonal forecast system based on a General Circulation Model to deliver relevant streamflow predictions. Additionally, the positive response to enhanced initial hydrological conditions pinpoints the efforts still needed to further improve land initialisation strategies, possibly through land data assimilation systems.

## 1 Introduction

The seasonal streamflow forecast (SSF) is an essential decision-making and planning tool for disaster prevention (e.g., floods and droughts), navigation, and water management applied to water supply, agriculture and hydropower generation (Clark et al., 2001; Hamlet et al., 2002; Chiew et al., 2003; Wood and Lettenmaier, 2006; Regonda et al., 2006; Luo and Wood, 2007; Kwon et al., 2009; Cherry et al., 2005; Viel et al., 2016). However, many regions lack operational forecast systems and dense streamflow/weather monitoring networks. To address this shortcoming, continental and global SSFs provide worldwide coverage of worthy prediction information (e.g., Crochemore et al., 2020; Emerton et al., 2018; Candogan Yossef et al., 2017; Pappenberger et al., 2013; Van Dijk et al., 2013).





Troin et al. (2021) propose a comprehensive classification of streamflow forecast systems into three groups based on the origin of the forcing: statistics-based streamflow prediction systems (SBSP), climatology-based ensemble streamflow prediction systems (ESP) and numerical weather prediction-based hydrological ensemble prediction systems (NWPB). SBSP approaches use historical streamflow or weather (or both) data to train a data-driven hydrological model, which, due to the absence of the physics to constrain it, requires long and continuous observational time series not always available (Troin et al., 2021). De-
spite statistical methods being the more widely developed and reliable methods in current operational forecast systems, their applicability can be limited because of the lack of physics description and robustness to represent future quick or long-term anthropogenic and climate changes (Candogan Yossef et al., 2017).

    ESP approaches (Day, 1985) use an ensemble of historical climate observations or (pseudo-)observations (such as satellite, radar and reanalysis of past weather data) to force one or more hydrological models (HMs). Most ESP multi-model studies
employ dynamical process-driven HMs rather than statistical data-driven HMs (Troin et al., 2021). Unlike SBSP, ESP can include physics representation in the HM, while past weather data only represents the climatology of the atmosphere without a link to the current initial state of the land or the atmosphere itself at the beginning of the forecast. Efforts to enhance the skill of the classical ESP include conditional weighting of the ESP ensemble members based on the El Niño–Southern Oscillation signal (Werner et al., 2004). While modified versions of ESP can improve streamflow predictions for shorter lead times,
their skill decreases faster over time compared to NWPB systems (Trambauer et al., 2015). To overcome this issue, model-based NWPB approaches propose using numerical weather prediction (NWP) systems or atmospheric predictions derived from global circulation models (GCMs) to yield ensemble atmospheric forecasts as inputs to the HM (e.g., Crochemore et al., 2017; Mendoza et al., 2017; Rosenberg et al., 2011).

    The seasonal streamflow forecast skill derives from the accuracy of the initial hydrological conditions (IHCs; of soil mois-
ture, groundwater, snowpack, and the current streamflow) and the future seasonal climate anomalies (FSCs; of temperature and precipitation) (Wood et al., 2016; Arnal et al., 2017; Yuan et al., 2015). As time progresses, the predictability of seasonal streamflow decreases, primarily due to the loss of memory in the IHCs and the increasing uncertainty in FSC predictions. The persistence of IHCs, depending on the season, catchment climate zone, and physiography, can extend from one to six months. Notably, the contribution of IHCs to predictability is more pronounced in arid and snowmelt-dominated hydroclimates
(Yuan et al., 2015; Shukla et al., 2013). Conversely, in regions dominated by rainfall, FCAs tend to significantly influence the predictability of seasonal streamflow (Wood et al., 2016). Forecasts entirely derived from the climatology of observed streamflow do not contain information on IHC and FSC since they are not initialised or atmospherically driven. Although atmospheric forcing in the ESP framework is climatology-based, introducing a hydrological model with IHCs constrains the forecast system and thus reduces the range of uncertainty. In NWPB approaches, FSC is simulated by a climate model, which
adds physics-based constraints to the system but may provide additional uncertainty in regions where it lacks skill. Therefore, it may be more straightforward to predict streamflow in large river basins with long-lasting IHCs (low IHCs uncertainty) and in regions with arid climates (lower rainfall FSCs uncertainty) (Wood and Lettenmaier, 2008; Shukla et al., 2013; Van Dijk et al., 2013; Yuan et al., 2015). In such cases, NWPB offers a more narrow ensemble than ESP methods (Wood et al., 2016; Li



et al., 2009). ESP is considered more reliable for long-range forecasting in regions where FSC dominates the other sources of
uncertainty, and NWPB fails to be skilful with respect to the long-term climatology (Demargne et al., 2014).

   Shortcomings inherent to land surface hydrological parameterisations and land surface initialisation of coupled GCMs have
discouraged the direct use of streamflow (or runoff) forecast products from these systems (Yuan et al., 2015). For this reason,
previous global scale studies based on dynamical methods rely on stand-alone hydrological models driven by bias-corrected
atmospheric forecasts from a GCM (Candogan Yossef et al., 2017; Emerton et al., 2018), in which explicit two-way mass
and energy feedback between land-atmosphere is not represented. However, coupled GCMs with consistent IHCs can produce
improved atmospheric seasonal forecasts in regions prone to a strong land-atmosphere coupling (Koster et al., 2004; Ardilouze
et al., 2017).

   On a global scale Candogan Yossef et al. (2017) suggest that the performance of the stand-alone approach, using the me-
teorological forecasts ECMWF S3, is close to that of the ESP forecasts. Such results, together with the recent evolution and
improvement of GCMs in terms of resolution, processes representation, hydrological parametrisation and land-surface initial-
isation, motivate the use of GCMs with embedded sophisticated river routing models (e.g., Decharme et al., 2019), to direct
production of seasonal streamflow forecasts.

   Thereby, we propose a global assessment of the SSF delivered by the Météo France operational forecast system SYS8,
based on CNRM-CM6-1 (Voldoire et al., 2019), an Atmosphere-Ocean General Circulation Model (AOGCM) embedding an
advanced river routing scheme coupled to the land-surface and atmosphere components, namely ISBA-CTRIP (Decharme
et al., 2019). To the best of our knowledge, the hydrological output of CNRM-CM6-1, initially developed by Centre National
de Recherches Météorologiques (CNRM) and Cerfacs for the sixth phase of the Coupled Model Intercomparison Project 6
(CMIP6, Eyring et al. (2016)), has never been evaluated in a forecasting configuration. The standard method to initialise the
CNRM-CM6-1 seasonal forecasts operational system is more advanced for ocean and atmosphere initial conditions than for
land initial conditions, given that the primary sources of seasonal predictability at the global scale originate from the ocean
(e.g., El Niño - Southern Oscillation). For this reason, we proceed to a 2-tier assessment of the impact of using an (i) online
coupled AOGCM-river rather than ESP and (ii) improving IHCs in the land-river components of the AOGCM. Here, the IHC
improvement is based on enhancing the representation of soil water content variability through the relaxation to a soil moisture
reanalysis specially developed for this study.

The following section presents an overview of the forecast systems and experimental design, as well as the observational
global streamflow database and forecast evaluation metrics. In the subsequent two sections, we address the impact of the IHCs
and the atmosphere-land-river coupling from global to basin scale to demonstrate the potential benefits of our approach. Finally,
we conclude with future scientific challenges and some final remarks.



## 2 Data and methods

### 2.1 Global forecast system

The Météo-France seasonal prediction system SYS8 (MF system 8; Batté et al., 2021) is based on the high-resolution version of the coupled CNRM-CM6-1 global climate model (Voldoire et al., 2019, 2017) used for CMIP6 (Eyring et al., 2016). It contributes to the seasonal forecast component of Copernicus Climate Change Services (C3S).

The streamflow forecast derives from the interaction between the atmosphere component ARPEGE-Climat 6.3 (Roehrig et al., 2020), the land surface component (ISBA), which simulates the runoff, and the advanced river routing (CTRIP), which simulates the streamflow river discharges (Decharme et al., 2019). In **ISBA**, the soil is discretised in 14 vertical layers, accounting for the soil hydraulic and thermal properties, while the multi-layer snow model simulates water and energy budgets separately in the soil and the snowpack. ISBA in one grid cell is tiled into 12 patches of soil and vegetation, which aggregates 500 land cover units at 1 km resolution present in the ECOCLIMAP-II database (Faroux et al., 2013), where mean seasonal cycles of snow-free albedo and leaf area index are prescribed from Moderate Resolution Imaging Spectroradiometer MODIS products at 1-km spatial resolution and the Normalised Difference Vegetation Index product from the SPOT/Vegetation. The soil textural properties (clay, sand, and soil organic carbon content) are given by the Harmonized World Soil Database (http://webarchive.iiasa.ac.at/Research/LUC/External-World-soil-database/HTML/) at a 1 km resolution. Topography is derived from the 1 km Global Multi-resolution Terrain Elevation Data 2010 (https://topotools.cr.usgs.gov/gmted_viewer/). Heterogeneities in precipitation, soil infiltration capacity, topography, and vegetation are considered through a comprehensive sub-grid hydrology scheme (Decharme and Douville, 2006; Decharme, 2007). In **CTRIP**, the result of rainfall excess, effective river-aquifer exchange, open water evaporation and inflow from the upstream cell is routed by a river model in which the streamflow velocity is solved dynamically via Manning's formula and assuming a rectangular river cross-section in a grid resolution of $0.5^o$ (Decharme et al., 2010).

CNRM-CM6-1 incorporates an explicit two-way coupling between ISBA and CTRIP *via* the SURFEX and OASIS-MCT interface (Voldoire et al., 2017). The coupling allows to consider (i) the dynamic river flooding in which floodplains interact with the soil and the atmosphere through infiltration, open-water evaporation, and precipitation interception (Decharme et al., 2012), while (ii) a two-dimensional diffusive groundwater scheme represents unconfined aquifers and upward capillarity fluxes into the superficial soil (Vergnes et al., 2014). The latter contributes to capturing active groundwater–river connections crucial to represent groundwater-sustained baseflow during dry seasons (Xie et al., 2024). More details on the model parametrisation and structure can be found in Decharme et al. (2019); Voldoire et al. (2019).

### 2.2 Experimental design

#### 2.2.1 Generation of land and river initial conditions

SYS8 derives land initial hydrologic conditions (IHCs) from a historical initialisation run, named ICL here, where the land-river component is unconstrained whereas the ocean and atmosphere are nudged towards the GLORYS12V1 (Lellouche et al.,





2021) and the ERA5 (Hersbach et al., 2020) reanalyses, respectively. We propose an enhanced initialisation run (ICL$_{nud}$) by relaxing soil moisture ($W_{soil}$) to fields obtained from a current $W_{soil}$ reconstruction. The soil moisture reconstruction was yielded through an offline land simulation (e.i., forcing the land-river components with ERA5 historical climate sequences). Then, the $W_{soil}$ of the historical initialisation run is nudged to the $W_{soil}$ from this reconstruction. The proposed IHC accounts

for an enhanced representation of soil moisture variability through (pseudo-)observed atmospheric forcing aiming to improve the forecast in basins where the initial soil water storage dominates the streamflow seasonal response.

The IHCs generated by ICL are applied to the benchmark forecast (Offline_ICL) and the online coupled SYS8 forecast (Online_ICL). The land-river component in the online system is also initialised with ICL$_{nud}$ to evaluate the impact on streamflow forecasting. Details of the model configurations and forcing are presented in Section 2.2.2.

**2.2.2   Forecast experiments**

Seasonal hindcast experiments were conducted for the three model configurations described below (see Table 1 and Figure 1).

- **Offline_ICL**: is the benchmark hindcast configured as the ESP classical approach. It is a land-river offline simulation initialised by the conventional initialisation run ICL.

- **Online_ICL** is produced by the online coupled system with conventional initialisation ICL.

- **Online_ICL$_{nud}$** is produced by the online coupled system with enhanced initialisation ICL$_{nud}$ based on a soil moisture reconstruction (SMR).

For each of the three forecast system configurations, we have generated two sets of hindcasts composed of 25 members, each one of them yielding a global four-month streamflow daily time series. The two sets were initialised on May 1st (JJA predictions) and November 1st (DJF predictions) between 1993 and 2017. The system Online_ICL is identical to the operational

SYS8 hindcast, except that for the latter, the ensemble is partly generated via a lagged initialisation method (e.g., Hoffman and Kalnay, 1983) while the ensemble members of Online_ICL (and Online_ICL$_{nud}$) stem from a burst initialisation, that is, all members have the same initialisation date.

To generate the benchmark hindcast Offline_ICL, the land-river model ISBA-CTRIP is forced by ERA5 historical climate (Figure 1) so that each year produces one of the 25 forecast members. We use leave-three-years-out cross-validation (L3OCV)

to select the forcing. In L3OCV, the year of the climate forcing cannot match the hindcast year nor the preceding year and the two following years to avoid artificially inflating the skill due to large-scale climate–streamflow dependence with influences lasting from seasons to years like the North Atlantic Oscillation (Dunstone et al., 2016). For example, to apply the L3OCV selection method to the hindcast of 1993, only forcing from 1996 to 2020 ensures 25 members. For the hindcast of 2000, only forcing from 1992 to 1998 and 2003 to 2020 is used. Unlike in the current hindcasting for validation, in operational forecast

systems, future climate information is not available; thus, only past climate information can be employed.

Before computing the forecast performance scores, the daily streamflow is averaged on a 3-month basis to represent the seasonal mean. The 3-month streamflow mean (JJA and DJF) is assessed across a global dataset of gauged basins with the





observational streamflow data described in Section 2.3. To localise the gauging stations in the correct grid pixel of the model river network, we applied a homemade methodology based on a distance and drainage area station-to-pixel comparison (see Munier and Decharme (2022) for more details and applications).

**Table 1.** Experiments configurations for land initial condition and hindcasts production (see Fig. 1).

| | Simulation | Initial condition | | | | Integration | | | |
|---|---|---|---|---|---|---|---|---|---|
| **ID** | **Description** | **Atm.** | **Ocean** | **Land** | **River** | **Atm.** | **Ocean** | **Land** | **River** |
| | | *Soil moisture reconstruction (SMR)* | | | | | | | |
| SMR | Offline land simulation to reconstruct soil moisture | Disabled | Disabled | Spin-up | Spin-up | Prescribed (ERA5) | Disabled | Free | Free |
| | | *Historical initialisation runs* | | | | | | | |
| ICL | Online coupling with Atm./Ocean nudged to reanalysis | ERA5 | Glorys | Spin-up | Spin-up | Nudged (ERA5) | Nudged (Glorys) | Free | Free |
| $ICL_{nud}$ | ICL nudged to own soil moisture reconstruction SMR | ERA5 | Glorys | Spin-up | Spin-up | Nudged (ERA5) | Nudged (Glorys) | Nudged (SMR) | Free |
| | | *Hindcasts* | | | | | | | |
| Offline_ICL | ESP Benchmark: offline with land initialisation from ICL | Disabled | Disabled | ICL | ICL | Prescribed* (ERA5) | Disabled | Free | Free |
| Online_ICL | Online with land initialisation from ICL | ERA5 | Glorys | ICL | ICL | Free | Free | Free | Free |
| $Online\_ICL_{nud}$ | Online with land initialisation from $ICL_{nud}$ | ERA5 | Glorys | $ICL_{nud}$ | $ICL_{nud}$ | Free | Free | Free | Free |

*The atmospheric ensemble forcing for the Offline_ICL hindcast is constructed from past climate years selected by a leave-three-years-out cross-validation procedure.






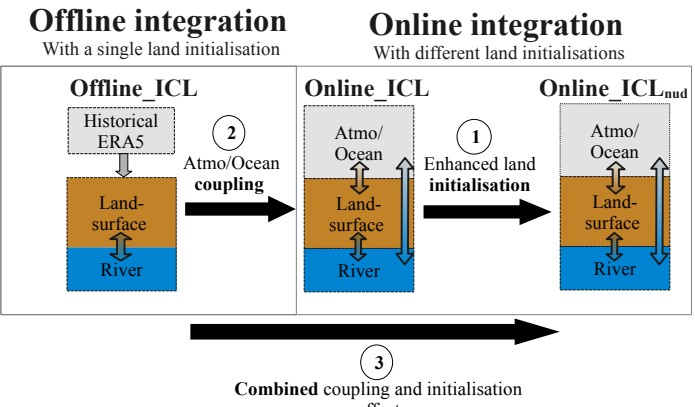

**Figure 1.** Comparison of model configurations. ICL: initial condition from the historical run with the online system; ICL$_{nud}$: initial conditions from a historical run with soil moisture relaxation to fields reconstructed from the offline land simulation SMR.





## 2.3 Streamflow observational database

Most previous works evaluate the "potential" streamflow predictability of a forecast system by adopting the perfect-model assumption, in which the streamflow forecast is compared to simulated streamflow (from a model driven by meteorological observations) instead of observed streamflow. Meanwhile, we compare the forecasts against observations because, in addition
to the IHC and FSC, it incorporates the uncertainty associated with model error (due to structure, physics, and parameter uncertainty) and provides actual (as opposed to potential) streamflow predictability, which is more valuable for end-users or the development of climate services. A database of 1755 flow gauge stations has been created, compiling the global streamflow open access datasets presented in Table 2. We have filtered the full dataset to remove stations with relatively small drainage areas poorly represented by the model resolution and those stations with more than $25\%$ of missing streamflow records in the
concerned season.

**Table 2.** Streamflow observed datasets.

| Dataset | Region | Reference |
| --- | --- | --- |
| GRDC: Global Runoff Data Centre | Global | http://www.bafg.de/GRDC/EN/Home/homepage_node.html |
| USGS: United States Geological Survey | United States | http://waterdata.usgs.gov/nwis/sw |
| HYDAT: National Water Data Archive | Canada | https://collaboration.cmc.ec.gc.ca/cmc/hydrometrics/www/ |
| French Hydro database | France | http://www.eaufrance.fr |
| Spanish Hydro database | Spain | http://ceh-flumen64.cedex.es/anuarioaforos/default.asp |
| HidroWeb | Brazil | http://www.snirh.gov.br/hidroweb/ |
| R-ArcticNet | Northern High Latitudes | http://www.r-arcticnet.sr.unh.edu/v4.0/AllData/index.html |
| Australian Bureau of Meteorology | Australia | http://www.bom.gov.au/metadata/19115/ANZCW0503900339 |
| China Hydrology Data Project | China | Henck et al. (2011) |
| HyBAm | Amazon basin | https://hybam.obs-mip.fr/ |

We conducted a correlation analysis to select the minimum drainage area considered in the study. For basins with an area higher than a certain threshold ($A_{threshold}$), Figure 2 shows the correlation between the basins area ($A_{basin}$) and the area estimated for the CTRIP routing model ($A_{CTRIP}$). With increasing $A_{threshold}$, the correlation increases, but the number of available basins reduces. The threshold is set to $6 \times 10^3$ km$^2$ (about two CTRIP cells per basin in mid-latitudes) to main-
tain a balance between the number of basins analysed and their geometrical representation and to avoid considering basins inside the spurious oscillating correlation curve (Figure 2). There are 1451 gauged basins with $A_{basin} \geq A_{threshold}$ with a $A_{basin}|A_{CTRIP}$ correlation of 0.9986.



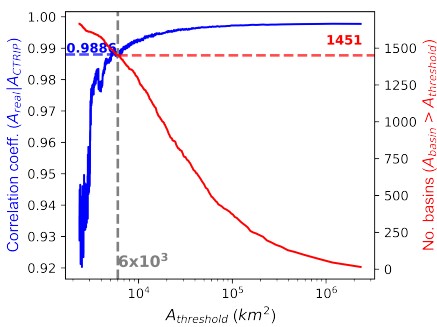

**Figure 2.** Correlation coefficient of $A_{real}|A_{CTRIP}$ and number of basins with area higher than a given $A_{threshold}$.

From the 1451 streamflow stations, we only consider those with less missing data than $25\%$ of the total data in the analysed season. Figure S1 in the Supplement shows the distribution, in spaces and frequency, of the full database and the selected
stations. The final dataset has 1071 stations in JJA and 1043 stations in DFJ, distributed in North America ($\approx 82\%$), Europe ($\approx 13\%$), South America ($3.5\%$), Africa ($1.7\%$), Asia and Australia ($0.4\% = 4$ stations). In Section 3.3, we remove 14 stations where some performance score magnitude exceeded the maximum machine number in double precision.

### 2.4  Streamflow bias correction

Typically, statistical post-processing methods are applied to compensate for errors in model structure or initial conditions,
correct biases, and improve ensemble dispersion (Troin et al., 2021). Such bias correction can be applied to atmospheric forecasts (such as precipitation, temperature, and evaporation) and/or to hydrological forecasts like runoff and streamflow (e.g., Petry et al., 2023; Tiwari et al., 2022; Gubler et al., 2020; Crochemore et al., 2016; Wood and Schaake, 2008). Bias correction of atmospheric forecast is often used in offline approaches where the hydrological model does not feedback on the atmosphere. However, our study uses an online atmosphere-ocean-land-river coupled model, for which bias correcting the atmospheric
forcing is irrelevant. Instead, we correct the streamflow forecast bias using the Empirical Quantile Mapping method (EQM) at each flow station. Unlike adjusting parametric distributions, the EQM method removes bias using empirical cumulative distribution functions (ECDFs) from observations and forecast percentiles. Roughly, the approach replaces the forecast values with observed values corresponding to the same non-exceedance probability (*i.e.* it calibrates the forecast distribution with the observed distribution by fitting the forecast values). Analogous to Tiwari et al. (2022), the bias-corrected streamflow $Q_c$ is
calculated as follows:

$$Q_c = F_o^{-1}[F_f(Q_f)] \tag{1}$$

where $F_f$ and $F_o$ are the ECDFs of forecast $Q_f$ and observation $Q_o$ streamflow, respectively.



## 2.5 Seasonal forecast assessment

Table 3 presents the deterministic and probabilistic scores used to evaluate the new forecast system performance. The thresholds
for the Brier score computation are based on the 3-month average of observed streamflow exceeded 66% (the lower tercile
Q66), 95% (Q95) and 10% (Q10) of the time. These thresholds characterise low, very low, and high flows (Liu et al., 2021).
The skill of the online approach is relative to the performance of the Offline_ICL benchmark.

     The significance of the precipitation correlation is calculated using the parametric Student t-test. All other significance
tests and confidence interval computations use the bootstrap approach, where 1000 random sub-samples are created from the
full sample to establish the probability distribution of the statistical estimator being analysed (e.g., the anomaly correlation
coefficient or the Kling-Gupta efficiency score). An estimator is considered significant if the p-value is less than or equal to
0.05.



**Table 3.** Performance scores used to assess and compare seasonal streamflow forecasting approaches.

| Notation | Name | Equation | Description |
|---|---|---|---|
| **Deterministic scores** | | | |
| Bias | Percent mean bias | $100 \times \dfrac{\sum(f_i - o_i)}{\sum o_i}$ | Range $(-\infty, \infty)$. It represents the average tendency of the forecast to underestimate or overestimate the observations, with 0 indicating that there is no bias. |
| RMSE | Root Mean Square Error | $\sqrt{\frac{1}{n}\sum(f_i - o_i)^2}$ | Range $[0, \infty)$. Lower values indicate better performance. |
| ACC | Anomaly Correlation Coefficient | $\dfrac{\sum(f_i - \bar{f})(o_i - \bar{o})}{\sqrt{\sum(f_i - \bar{f})^2 \sum(o_i - \bar{o})^2}}$ | Range $[-1, 1]$, with perfect score of 1. It measures the linear association between forecasts and observations (or pseudo-observations). |
| KGE | Kling-Gupta Efficiency Score | $1 - \sqrt{(ACC - 1)^2 + (DQR - 1)^2 + (QR - 1)^2}$ | Range $(-\infty, 1]$, with 1 being the optimal value. It considers correlation, bias, and variability error. |
| **Probabilistic scores** | | | |
| BS | Brier Score | $\frac{1}{N}\sum_{i=1}^{N}((1 - F_f(Q_{thr})) - \mathcal{H}'(o_i - Q_{thr}))^2$ | Range $[0, 1]$, where lower values indicate better and sharper forecasts. Measures the accuracy of probabilistic predictions and the bias in the probability space. |
| CRPS | Continuous Ranked Probability Score | $\frac{1}{N}\sum_{i=1}^{N}\int_{-\infty}^{\infty}(F_f(f_i) - \mathcal{H}(o_i - f_i))^2 df_i$ | Range $[0, \infty]$. Quadratic difference between the cumulative distribution function (CDF) of an ensemble forecast and the empirical CDF of the observation. Lower values indicate better performance. |
| **Generic Skill score** | | | |
| ABS | Absolute Skill Score | $\|\text{Score}_{\text{offline}} - \text{Score}_{\text{perfect}}\| - \|\text{Score}_{\text{online}} - \text{Score}_{\text{perfect}}\|$ | ABS ranges $(-\infty, 1]$ and RES ranges $(-\infty, \infty)$. It compares the current online system forecast against the offline reference forecast. perfect skill: RES $= 1$ (ABS $= \|\text{Sc}_{\text{off}} - \text{Sc}_{\text{perf}}\|$). no skill: RES $= 0$ (ABS $= 0$). skill degradation:RES $< 0$ (ABS $< 0$). **Note:** Any deterministic or probabilistic score can be used. ABS/RES is the magnitude/fraction of the score improvement (or degradation for negative values). |
| RES | Relative skill score | $1 - \dfrac{\text{Score}_{\text{online}} - \text{Score}_{\text{perfect}}}{\text{Score}_{\text{offline}} - \text{Score}_{\text{perfect}}}$ | |

$N$: Total number of forecasts; $f_i$: Forecast 3-months ensemble mean for year $i$; $o_i$: Observation 3-months mean for year $i$; $\bar{f}$: Temporal average over forecast ensemble means; $\bar{o}$: Temporal average of observations; $DQR = \frac{S_f}{S_o}$: forecast-to-observation standard deviation ratio; $QR = \frac{\bar{f}}{\bar{o}}$: forecast-to-observation mean ratio; $Q_{thr}$ is a threshold that represents the occurrence of a hydrological event; the step function $\mathcal{H}'(o_i - Q_{thr})$ is zero if $o_i \leq Q_{thr}$ or one otherwise; $F_f(f_i)$: Cumulative distribution function of ensemble forecast; the Heaviside step function $\mathcal{H}(o_i - f_i)$ is zero if $f_i < o_i$ or one if $f_i \geq o_i$; $\text{Score}_{\text{offline}}$: Score of Offline_ICL benchmark reference forecast; $\text{Score}_{\text{perfect}}$ score of a perfect forecast.



# 3 Results

The first two sub-sections explore the performance of the two primary factors of hydrologic predictability, namely the initial
hydrologic conditions (Section 3.1) and the future climate seasonal anomalies (Section 3.2). Section 3.3 presents the evaluation
of the seasonal prediction skill to highlight the joint and separate impacts of the coupling and the enhanced land initialisation.

## 3.1 Initial hydrologic conditions

We assess the global performance of the river streamflow simulated by the initialisation runs (ICL and $ICL_{nud}$) against historical
streamflow observations. For this purpose, we compare the initial-month mean streamflow (May for JJA and November for
DJF) against the observed one over the 1993-2017 period. Figure 3 presents three performance metrics of the comparison

**Figure 3.** Comparison between May streamflow mean of initialisation run against the observed one over 1993-2017. Left column: ICL bias
(a), root mean square error (mm/d) (d), and anomaly correlation (g). Middle column: difference with the $ICL_{nud}$ enhanced land initialisation
Bias (b), root mean square error (mm/d) (e), and anomaly correlation (h). Right column: distribution of Bias for each experiment (c),
accumulated distributions of the root mean square (f), and anomaly correlation (i).



(BIAS, RMSE and ACC). Note that only stations with less than 25% of missing data during the corresponding month are considered in the following analysis.

**Figure 4.** Comparison between November streamflow mean of initialisation run against the observed one over 1993-2017. Left column: ICL Bias (a), root mean square error (mm/d) (d), anomaly correlation (g). Middle column: difference with the $ICL_{nud}$ enhanced land initialisation Bias (b), root mean square error (mm/d) (e), anomaly correlation (h). Right column: distribution of Bias for each experiment (c), accumulated distributions of the root mean square (f), and anomaly correlation (i).

For May, the streamflow Bias of ICL tends to be positive in the dryest regions (Fig. 3a). Overall, the $ICL_{nud}$ benefits the negative Bias reduction, while positive biases are worsened (Figure 3b-c). Besides, the RMSE is generally smaller with $ICL_{nud}$, in particular over regions with large RMSEs in ICL (Figure 3d-f). In seasonal forecasts, the temporal correlation between

the forecasted and the observed anomalies is crucial since it indicates the capability of capturing the inter-annual variability of streamflow departures from the mean value. The spatial distribution of the difference in anomaly correlation coefficient $|ACC_{ICL}-1|-|ACC_{ICL_{nud}}-1|$ in Fig. 3h shows that the soil moisture nudging improves the temporal dynamics of the simulated





streamflow in May over most of the 1067 gauging stations. The result is verified in the Fig. 3i, which reports up to 20% more stations with $ACC > (0.4 - 0.6)$.

The performance of the river initialisation in November (used for DJF forecasts) is presented in Fig. 4. $ICL_{nud}$ tends to reduce the mean Bias of stations displaying a high positive Bias in ICL, more frequently located over dry regions (Figure 4a-b). The Bias global distribution in Fig. 4c confirms a reduction of high positive Bias, favouring the concentration of Bias values closer to zero than ICL. However, unlike JJA, in DJF, $ICL_{nud}$ induces more stations with higher RMSE and lower ACC. In Section 3.3, we show and discuss the impact of the initial hydrologic condition (IHC) degradation on the hindcasts in boreal winter.

## 3.2    Precipitation and temperature skill

One way to bring out the influence of the land-atmosphere coupling is to assess the impact of different land IHCs on the atmospheric forecast. The performance of the atmospheric seasonal forecast is presented in Figures 5 and 6, in particular, for two of the most important water cycle drivers: precipitation and near-surface temperature. Precipitation is compared against the Multi-Source Weighted-Ensemble Precipitation (MSWEP v2, Beck et al. (2019)) and the temperature against the Climatic Research Unit gridded Time Series (CRU TS v4.05, Harris et al. (2020)).

A global view does not reveal marked changes in terms of ACC for the atmospheric predictions. However, from a continental to regional view, differences are noticeable. In the boreal summer (Figure 5), the enhanced initialisation $ICL_{nud}$ tends to increase the precipitation correlation in latitudes south of $20°S$, including the Parana basin, with a degradation on the northeast of Brazil and north of $40°N$. Between latitudes $30°N - 50°N$, the case of Europe is worth pointing out, with improved precipitation prediction. The temperature is less sensible to the land initialisation in summer, but degradation is more concentrated in higher latitudes (north of $40°N$ and south of $20°S$). In winter, the regions with precipitation and temperature performance decrease are concentrated in North Africa, Europe and Asia (Figure 6). We have shown evidence of the impact of land IHC on the performance of seasonal atmospheric forecasts as proof of the importance of the land-atmosphere feedback. In the following section, we will explore the sensitivity of the SSF to enhanced IHCs in a fully coupled global forecast system.

## 3.3    Impact of initialisation and coupling on streamflow forecast skill

The hindcast performance of the ESP benchmark (Offline_ICL) is compared against the hindcasts of the fully coupled configurations with two different land initialisations (Online_ICL and Online_$ICL_{nud}$) to determine the contributions of initialisation and land-atmosphere coupling. Unlike online configurations, where the model forecasts the atmosphere, in Offline_ICL, the atmosphere forcing is based on climatology without land-atmosphere feedback. More details on the three configurations can be found in Section 2.2.2.

### 3.3.1    Global view

In summer, the spatial distribution of the anomaly correlation coefficient of the hindcasts compared to the benchmark Offline_ICL reveals a limited effect of the coupling Online_ICL (Figures 7a-b). However, a substantial improvement of the JJA





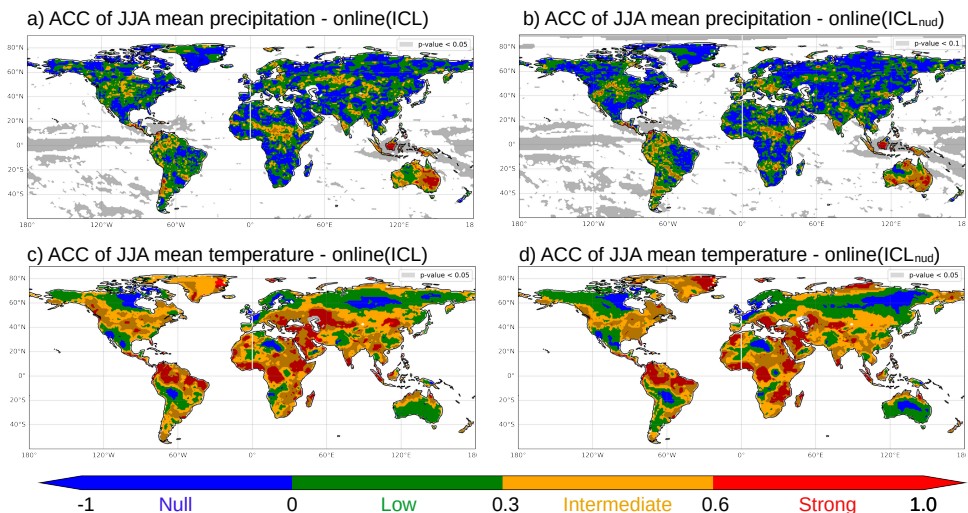

**Figure 5.** Comparison of Online_ICL and Online_ICL$_{nud}$ atmospheric forecasts for the anomalies correlation coefficient of the JJA 3-month mean precipitation (a and b) and temperature (c and d).

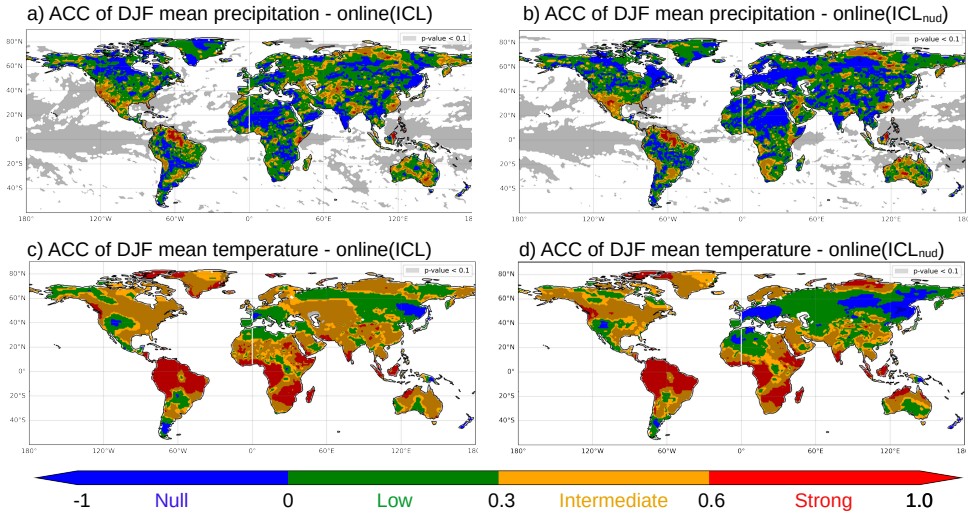

**Figure 6.** Comparison of Online_ICL and Online_ICL$_{nud}$ atmospheric forecasts for the anomalies correlation coefficient of the DJF 3-month mean precipitation (a and b) and temperature (c and d).

streamflow forecast is achieved with the enhanced initialisation Online_ICL$_{nud}$ (Figures 7c), also drawn by the cumulative 250 distribution in Fig. 7d.





For winter, in the second column of Fig. 7, the coupled hindcasts with both land initialisations yield a remarkable increase of stations with intermediate and high correlation. The cumulative distribution of the ACC, in Fig. 7d, confirms that the number of stations with an ACC greater than 0.5 (0.7) increases to more than 25%(7%). In addition, from Online_ICL$_\text{nud}$ to Online_ICL, the ACC is slightly reduced, especially for basin outlets in the north of $40°N$. It suggests that soil moisture treatment in ICL$_\text{nud}$

tends to reduce the ability of the system to predict winter streamflow dynamics in basins with strong ice influence.

A global view of the impact of bias correction, coupling, and enhanced initialisation is presented in Figure 8. For the three set-up models, the cumulative distributions ACC and KGE are computed for the raw and bias-corrected hindcasts for the boreal summer (JJA) and winter (DJF) seasons. Before and after bias correction, both online hindcasts outperformed offline configuration. Furthermore, both metrics confirm that the hindcast skill with the enhanced initial condition ICL$_\text{nud}$ is improved

compared to Online_ICL in summer but slightly worsened in winter. As expected, the ACC was weakly modified after bias correction ( Figure 8a to b), while the number of stations with positive KGE increased up to $20\%$, excepting the Offline_ICL that was less sensitive to the bias correction in DJF (Figure 8c against d). It states that the biggest contribution of bias correction comes from the forecasted-to-observed streamflow mean and standard deviation ratio.

Figure 9 displays the deterministic and probabilistic scores as a function of the basin area. For all JJA streamflow predictions,

the KGE (and its component scores in Fig. 9a) and the CRPS (Figure 9b) reveal an improvement with increasing drainage area, while low, mean and high flow predictions (BS95, BS66, and BS10) report weak basin area dependence. The figure confirms that Online_ICL$_\text{nud}$ outperforms Online_ICL for both deterministic and probabilistic metrics. Unlike Offline_ICL, the median scores of coupled systems show weak to null dependence on basin area in winter, while the amplitude of the variation decreases with the area (Figures 9c and d). Besides, in winter, the Offline_ICL produces poor-quality forecasts in most gauge stations,

as reported by the low median KGE and ACC values. However, for basins with a drainage area $\geq 10^6$ km$^2$, the Offline_ICL is close to Online_ICL and Online_ICL$_\text{nud}$. It should be noted that Online_ICL$_\text{nud}$ has a negative impact in winter, reducing the mean forecast performance for the basin area ranges, in terms of variability (ACC in Figure 7) and oscillation amplitude ($\frac{S_f}{S_o}$: forecast-to-observations deviation ratio in Fig. 9c).

The density, quantity, and distribution of the flow gauge stations vary significantly between continents. As shown in Fig-

ure 7d, the distribution of the scores in the frequency space tends to reflect the continent with more gauge stations, such as North America in this case. Therefore, in the next section, we will assess the forecast performance on a continental scale.



**Figure 7.** ACCs of bias-corrected streamflow hindcasts computed against observations in JJA (first column) and DJF (second column). Offline_ICL benchmark (First row) and the online coupled configurations with conventional initialisation (second row) and improved initialisation (third row). Cumulative distribution of the anomaly correlation coefficient of the corresponding season (last row). Markers with transparency represent stations with a statistically non-significant ACC at the 95% confidence level.





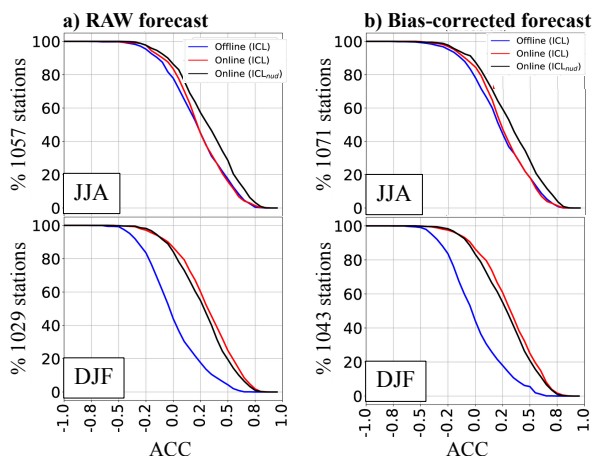

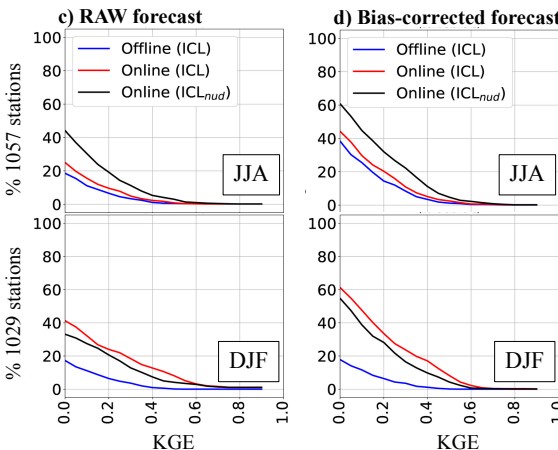

**Figure 8.** Cumulative global distributions of anomaly correlation coefficient (left panel) and Kling-Gupta Efficiency Score (right panel) of raw (a and c) and bias-corrected (b and d) forecasts.



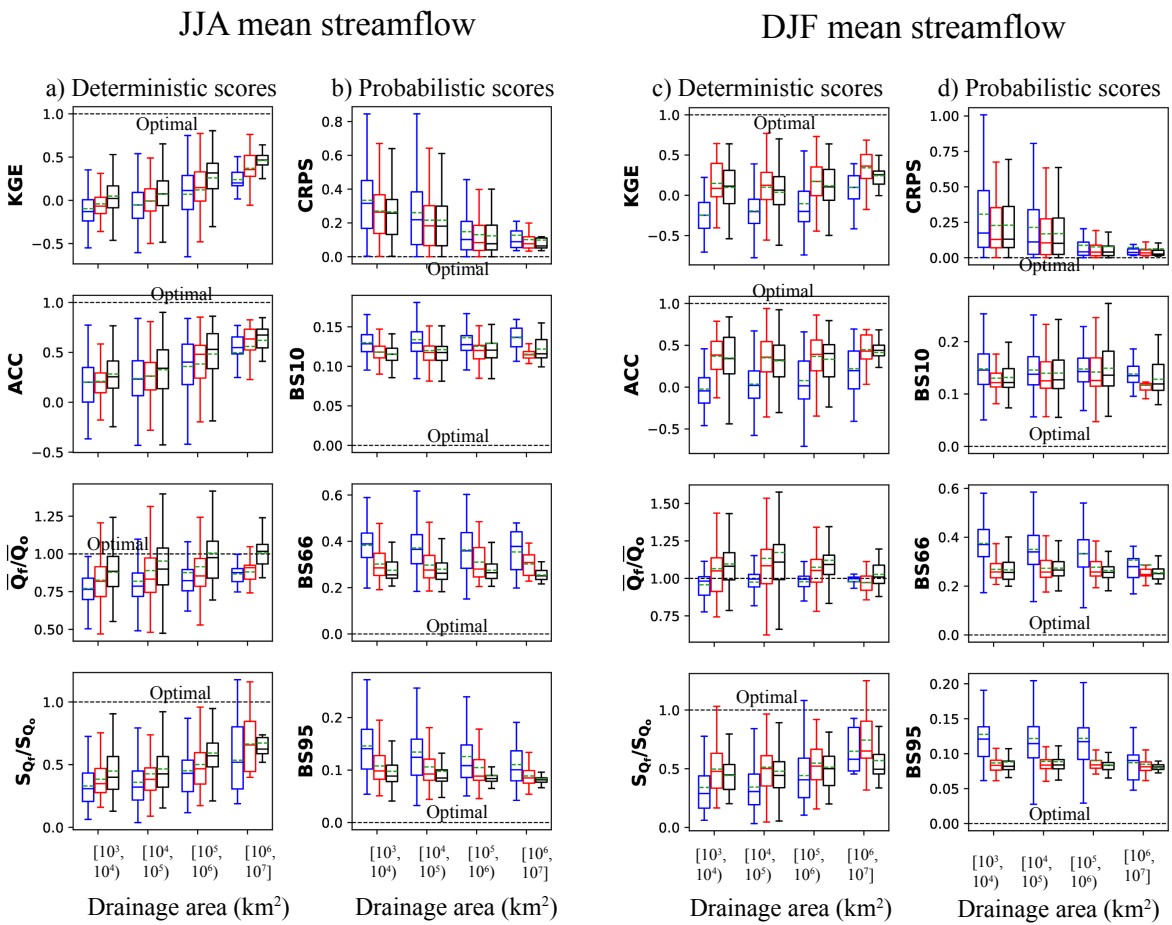

**Figure 9.** Performance of the hindcasts 3-month mean streamflow as a function of the basin area for summer (two left columns) and winter (two right columns). Scores computed in all gauging stations of the global database are visualised in box plots of deterministic (a and c) and probabilistic (b and d) scores. Four basin area classes are defined with (131, 670, 215 and 41) stations for JJA and (123, 656, 210 and 40) in DJF. The colour of the box represents the model configuration: – Offline_ICL, – Online_ICL or – Online_ICL$_{nud}$. The continuous line in the box is the median, while the dashed green line indicates the mean value.





### 3.3.2 Continental view

Figures 10 and 11 present the KGE spatial and frequency distribution for summer and winter in North America, Europe, South America and Africa. In summer, on the seasonal time scale, the river discharge tends to be driven by water released from the
basin water storage. Consequently, the initialisation of the soil water content (soil moisture) of the land component plays a major role in streamflow prediction. This claim applies to North America and Europe, where enhanced land initialisation has a more positive impact than atmospheric coupling with conventional initialisation. Meanwhile, in winter, streamflow is primarily driven by precipitation. This means that rainfall forecasts matter more than water content and land initialisation quality. The cumulative KGE distribution of Fig. 10 confirms that, independently of the initialisation, the atmospheric forecast coupled with
land yields improved predictions with respect to the Offline_ICL.

The greatest improvement in South American rivers for both seasons comes from the dynamic atmospheric forecast incorporated in the coupled systems. Due to the few gauge stations in Africa, the cumulative distribution does not provide robust information. As a result, the different levels of coupling and initialisation do not show evidence of impact on the seasonal prediction of streamflow in the 15-17 gauging stations evaluated in Africa. However, unlike the DJF season, all the model setups
provide satisfactory predictions in JJA.

Before advancing in the skill analysis, we have identified basins exhibiting pertinent hindcasts accuracy, whereby at least one of the three hindcasts configurations yields a significant positive anomaly correlation coefficient (indicated by the lower $95\%$ confidence bound of ACC being negative). This screening retains 650 stations in JJA and 620 in DJF (Figure S2), presenting a distribution of drainage areas, as depicted in the initial data (Figure S1), predominantly skewed towards values below $2 \times 10^5$
$\text{km}^2$, with a substantial number of basins exceeding $10^6 \text{ km}^2$ in area.

In addition to the ACCs for the comprehensive dataset in Figure 7, Figure S3 presents the ACC map of the benchmark alongside the absolute skill score of online configurations after the exclusion of stations exhibiting negative correlation hindcasts across all configurations.

The anomaly correlation relative skill of online approaches with different initialisations compared to the Offline_ICL is
presented in maps and cumulative distributions of Figure 12. During summer, in North America's arid regions (Figure 12a-b), the enhanced initialisation provides about $25\%$ of additional skill (Figure 12c). However, in winter, it degrades the forecast mainly in latitudes $> 60° N$ (Figure 12d-e) in about $9\%$ (Figure 12f). In South America, the ACC skill increases in summer by about $15\%$ because of the initialisation, while in winter, it yields a close to $1\%$ slight degradation.

To summarise the SYS8 assessment, the relative skill of online approaches is presented for three deterministic and four
probabilistic scores in Figure 13. A positive relative skill score indicates an improvement with respect to Offline_ICL, where 1 corresponds to the perfect score. For example, in the North American winter season, the median ACC is $45\%$ (for Online_ICL) and $38\%$ (for Online_ICL$_{nud}$) closer to the perfect correlation than Offline_ICL. For boreal summer (top row of Fig. 13), all skill metrics confirm the added value of enhanced land initialisation ICL$_{nud}$ to improve streamflow forecasts. However, over South America and Africa, the probabilistic metrics show more elusive improvement (or degradation). For boreal winter
(bottom row of Fig. 13), forecasts show higher performance than the benchmark in general but suggest a minor impact of







**Figure 10.** Comparison of seasonal streamflow hindcasts performance in boreal summer JJA. The maps display the station-wise KGEs for the seasonal streamflow of the (a) Offline_ICL hindcast and the absolute skill score of (b) Online_ICL and (c) Online_ICL$_{nud}$ experiments. Column (d) exhibits the 3 KGE cumulative distributions for the corresponding continent. Markers with transparency represent stations where KGE is significantly negative with confidence of 95%.



**Figure 11.** KGEs for the streamflow in boreal winter of the (a) Offline_ICL hindcast, and the absolute skill score of (b) Online_ICL and (c) Online_ICL$_{nud}$ experiments. (d) KGE cumulative distributions for the corresponding continent. Markers with transparency represent stations where KGE is significantly negative with a confidence of 95%.

improved land initial conditions. The deterministic scores also reveal that the skill gain for online approaches is sharper for ACC than RMSE and KGE, which denotes a better ability of coupled forecast systems to capture the interannual variability of



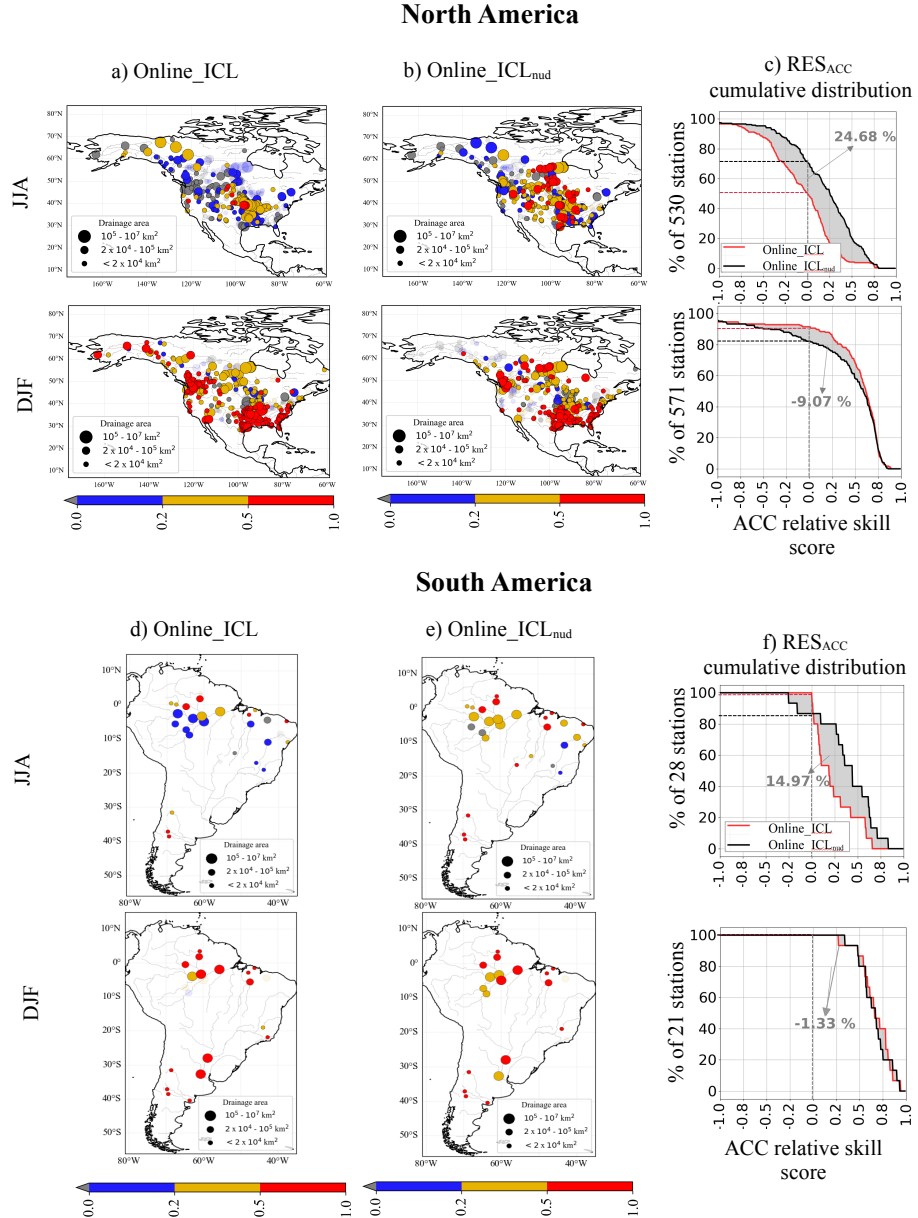

**Figure 12.** Comparison of online hindcasts with respect to the offline benchmark reference in North America (upper panel) and South America (lower panel). The relative skill score RES of anomalies correlation is $1 - (ACC_{\text{online}} - 1)/(ACC_{\text{offline}} - 1)$. In each panel, the two left columns present the RES map, and the right column presents its cumulative distribution for summer (first row) and winter (second row). The grey area between cumulative distribution curves is the percentage of ACC skill added by the new initialisation in relation to the conventional one (negative values indicate skill degradation).





river streamflows. In JJA and DJF, the reduction of RMSE, if any, remains limited for most conrespecttinents. Additionally, the Brier skill score for high flows BS10 is generally lower than the BS66 and BS95 for mean and low flows. This result suggests

either that the forecast systems anticipate better seasonal droughts than excessive cumulated precipitation, or that dry initial conditions, associated to low flows are more persistent than the wet counterpart.

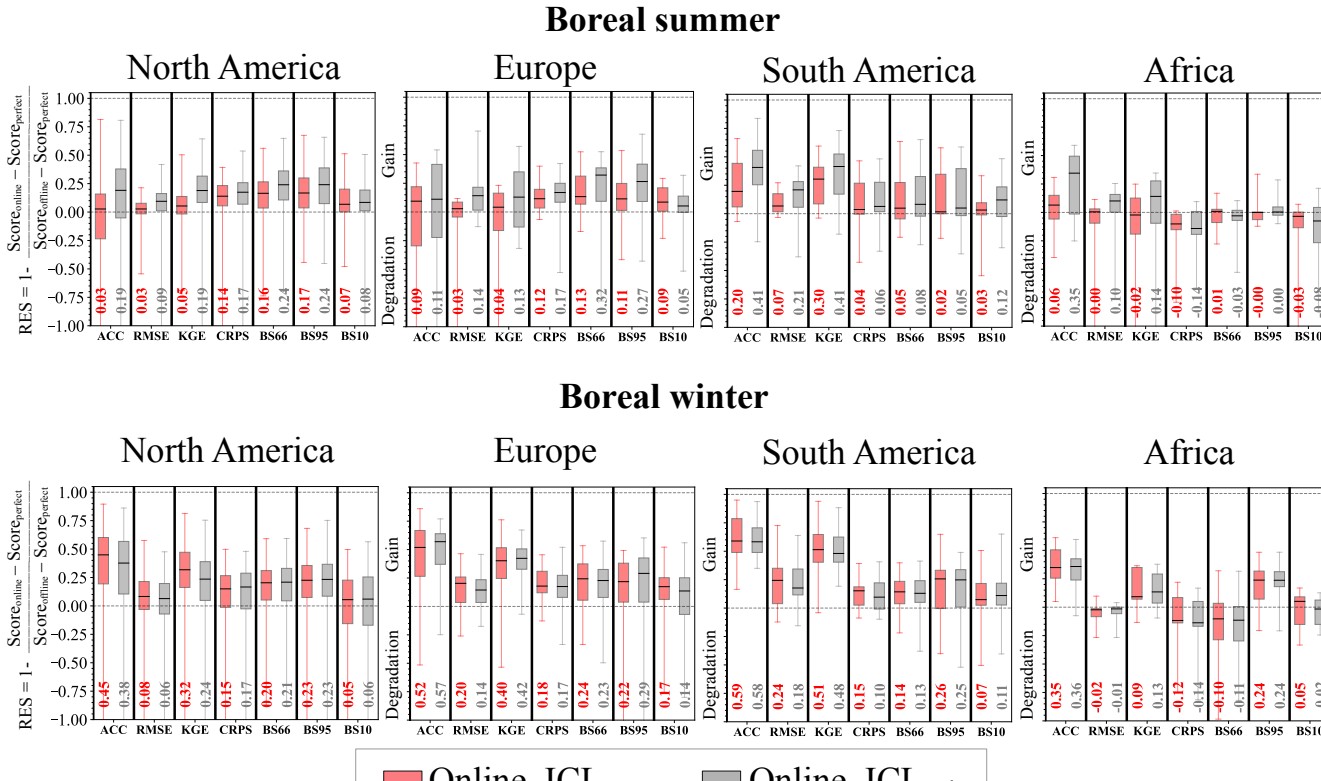

**Figure 13.** Distribution of relative skill metrics for the stations of each region with respect to the benchmark Offline_ICL for Online_ICL (red box-plots) and Online_ICL$_{nud}$ (grey box-plots) in JJA (top row) and DJF (bottom row). Colored numbers indicate the median values.

### 3.4 When, where and why the SYS8 is skilful

The enhanced land-river initialisation was designed to capture soil moisture spatial and temporal dynamics and thus improve water storage variation. For this purpose, soil moisture is relaxed to reconstructed fields of a reanalysis based on a surface

model driven by ERA5 atmospheric forcing.

In boreal summer, enhanced land initialisation is more critical than using a fully coupled GCM-derived forecast system, since only the former approach led to improved forecasts. For this season, the highest impact of initialisation on forecast skill





occurs in large basins of semi-arid regions, which are strongly sustained by the water storage naturally released during low flows.

In boreal winter, the new IHC negatively affected basins in high latitudes, probably due to the potential disruption of the energy and ice-liquid water budget in the soil induced by the lack of nudging of soil temperature (e.g., Ardilouze and Boone, 2024), which could lead to spurious model adjustment through excessive or reduced runoff. Confirming this would deserve a dedicated evaluation beyond the scope of this study. In South America, the coupling is beneficial (for both IHCs) in JJA and DJF, which is consistent and directly related to the strong ACC of precipitation and temperature provided by the online systems

in most of the basins analysed in South America (see Figures 6 and 5). For Africa, no robust conclusions can be drawn from the results due to the reduced sample of stations. Still, predictions from all model configurations in DJF were poor, with fewer than 40% stations exhibiting a positive KGE.

## 4   Conclusions

In this paper, we assess the Météo France global streamflow seasonal forecast operational system (SYS8) based on the latest
version of the CNRM global climate model CNRM-CM6-1. This model incorporates an advanced river routing model that interacts with the land surface component *via* superficial/sub-superficial runoff, unconfined aquifers water exchange and saturated floodplain (re)infiltration, and with the atmosphere through free-water evaporation and precipitation interception on floodplains. We thus employ SYS8 to produce a 25-member ensemble daily streamflow hindcasts extending up to 4 months, with burst initialisation on May 1st and November 1st, to predict respectively the boreal summer and winter global seasonal
streamflow from 1993 to 2017.

    The seasonal streamflow anomalies are evaluated against observations to asses the actual skill with respect to a classical Ensemble Streamflow Prediction (ESP) offline approach, used as a benchmark forecast. In addition to assessing the skill of the coupled forecast system, we compare two different land initialisation strategies. We found that the seasonal streamflow forecast (SSF) of SYS8 can be skilful during the summer and winter boreal seasons.

The main novelty and conclusions of this work can be condensed into the key points listed below.

- Our results demonstrate, for the first time, the potential to utilise direct global streamflow forecasts issued by a global climate model fully coupled with a river-floodplain model. The convenient single-step workflow natural of the coupled approach employed in SYS8 allows simultaneous production of atmospheric and streamflow forecasts, while the online coupled model ensures consistency in conservation laws at the initialisation and during the forecasting.

- In boreal summer, the water storage initialisation has the largest positive impact on the SSF quality. The improvement is sharper in dry and semi-arid regions and for the largest basins where high storage capacity drives the basin response during low flow periods typical of summer.

- In boreal winter, the streamflow variability tends to be mostly induced by precipitation seasonal anomalies, thereby reducing the impact of the initialisation on the SSF performance. The atmospheric predictive capacity of the coupled

model, albeit relatively limited over mid-latitude regions, leads to SSF overall being more accurate than the benchmark ESP offline forecast driven by climatology-based atmospheric forcing.

Current efforts to extend the assessment of the SYS8's actual skill to other parts of the globe include augmenting the observation database with discharge time series from regional and local flow station datasets. In future work, we will also evaluate the system's potential predictability using the perfect model approach (i.e., employing the river streamflow time series

from the CTRIP output of the initialisation run as "pseudo-observations"), which allows for system assessment in virtual stations equally distributed across the globe.

Our study also provides insights for improving the forthcoming generation of forecast systems for hydrological predictions, particularly regarding initialisation methods.

In order to further confirm the findings of this work and explore other seasons, we are developing a novel and more robust

land-river initialisation strategy in the framework of the Horizon Europe project CERISE *via* the in-house global land data assimilation system LDAS-monde (Albergel et al., 2020). Finally, in the longer term, we expect forecast improvements from the high-resolution version of CTRIP (Munier and Decharme, 2022) to replace the current coarser model, along with the activation of a novel irrigation scheme (Decharme et al., 2024, personal communication) in the next-generation of the CNRM-CM GCM for CMIP7.

*Author contributions.* Narváez-Campo G. and Ardilouze C. conceived, planned and carried out the experiments. Narváez-Campo G. took the lead in writing the manuscript and developed the interactive post-processing tool used to interpret the results. Ardilouze C. provided critical feedback and helped shape the research, analysis and manuscript.

*Competing interests.* Authors declare that no competing interests are present.

*Acknowledgements.* This work is part of the CERISE project (Grant Agreement No. 101082139) funded by the European Union. The authors

thank Dr. Simon Munier for providing the CTRIP river discharge evaluation tool and the aggregated observational database.



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
