# Peer review of "Skilful Seasonal Streamflow Forecasting Using a Fully Coupled Global Climate Model"

_EGUsphere, 2024_

## Author Response (AR1)

**Response to the referee #1 of the research article egusphere-2024-2962: Skilful Seasonal Streamflow Forecasting Using a Fully Coupled Global Climate Model**

Gabriel Fernando Narváez-Campo, Constantin Ardilouze

April 23, 2025

We thank the referee for the feedback. We will prepare a revised manuscript addressing the comments. We have organized this reply document as follows:

- The referee comments are in black.

- Our responses are in blue.

- The additions/modifications proposed for the manuscript revision are in red.

- Figures prepared for the reply have the prefix "R", such as Figure R1.

**Revisions**

1. The author evaluated summer and winter streamflow forecast one month in advance, I wonder whether two or even three lead months are considered (not 3-month mean)? Are the results consistent with the conclusions in the manuscript, or are the results of the online coupled forecast system used superior to the ESP, as the authors state in the introduction, "while modified versions of ESP can improve streamflow predictions for shorter lead times, their skill decreases faster over time compared to NWPB systems", maybe for longer lead times, the effectiveness of online coupled forecast system in predicting seasonal streamflow improves more.

   We understand the comment regarding the lead time skill dependence and thank the referee for the opportunity to clarify these key points in detail in the following and the revised version of the manuscript. We demonstrated that the discharge 3-month mean excluding the initial month (see Fig. R1) is better predicted by the coupled forecast system (e.g., see Figures 7d and 7h of the manuscript). However, the effect of lead time has not been explicitly discussed. To address this, we have performed a monthly lead time analysis to verify that our findings regarding the 3-month mean are consistent for a 1-month analysis, taking lead time dependence into account.

[Figure]

**Figure R1.** Global performance of forecasting systems for summer JJA and winter DJF seasons (discharge 3-month mean). Cumulative frequency distribution of anomaly correlation coefficient of hindcasts, for 1993-2017. (Same Figures 7h and 7d of the paper).

Figure S4 shows the cumulative frequency distributions of ACC for each of the four months forecasted. From the figure, we can remark on the following points.

- As expected, the performance decreases with lead time in all the forecast system configurations.
- In boreal summer JJA, the hindcast with improved land initialisation Online_ICL$_{nud}$ outperforms Offline_ICL benchmark at all lead time months. Conversely, Online_ICL is better than Offline_ICL for lead times over one month, as indicated by the slightly higher number of stations with positive but low correlations (about ACC<0.3).
- In boreal winter DJF, Online_ICL and Online_ICL$_{nud}$ performance is similar but always better than the Offline_ICL. Online_ICL is slightly better than Online_ICL$_{nud}$ in January (for 0.2<ACC<0.6) and February predictions (0.1<ACC<0.3).

The remarks are consistent with the conclusions derived from the 3-month discharge mean presented in the manuscript.

[Figure]

**Figure S4.** Global performance of forecasting systems at different lead times for summer JJA and winter DJF boreal seasons. The anomaly correlation coefficient for 1993-2017.

To strengthen our conclusions derived from the 3-month discharge mean, we will briefly analyse the monthly lead times in the manuscript, analogous to what is presented here, with additional figures located in the supplementary material.

2. Figure 1 demonstrates comparison of model configurations, but it is not sufficiently intuitive to understand and what the numbers in the figure represent is not explained. Also, "to generate the benchmark hindcast Offline_ICL, the land-river model ISBA-CTRIP is forced by ERA5 historical climate (Figure 1) so that each year produces one of the 25 forecast members" (Line 143), why does each year produce one of the 25 forecast members? The author mentioned 25 members several times, what specifically does members refer to?

We thank the referee for the comments on improving the article's readability. To include the considerations in section *2.2.2 Forecast experiments*, we propose enhancing Figure 1 (and caption) to include the configuration details more clearly while rewriting the referred paragraph to enhance the readability.

To generate the benchmark hindcast Offline_ICL, the land-river model ISBA-CTRIP is forced by ERA5 historical climate (Figure 1) so that each year produces one of the 25 atmospheric forecast members. We use leave-three-years-out cross-validation (L3OCV) to select the forcing. In L3OCV, the year of the climate forcing cannot match the hindcast year nor the preceding year and the two following years to avoid artificially inflating the skill due to large-scale climate–streamflow dependence with influences lasting from seasons to years like the North Atlantic Oscillation (Dunstone et al., 2016). For example, to apply the L3OCV selection method to the hindcast of 1993, forcing of years 1991 and 1996-2019 ensures 25 members. For the hindcast of 2000, forcing from 1991 to 1998 and 2003 to 2019 is used. Unlike in the current hindcasting for validation, in operational forecast systems based on the ESP Offline approach, future climate information is unavailable; thus, only past climate information can be employed.

[Figure]

**Figure 1.** Schematic of offline and online forecast system configurations and corresponding land-river initialisations. ICL: initial condition from the historical run with the online system; ICL$_{nud}$: initial conditions from a historical run with soil moisture relaxation to fields reconstructed from the offline land simulation SMR. As illustrated by the grey-filled arrows, the design of the experiment allows the evaluation of the coupling effect, the initialisation effect or both.

3. In chapter 3.2, the author shows the performance of the atmospheric seasonal forecast is presented in Figures 5 and 6, in particular, precipitation and near-surface temperature. Please highlight in the figures where the author mentioned in the paragraph. In Figure 5, the ACC of global precipitation is overall lower in Online_ICLnud than in Online_ICL in summer, especially in South America and Australia, also Online_ICLnud has more blue areas than Online_ICL in winter that means more negative ACC of precipitation and near-surface temperature. Can the author give some explanations?

From a global view, we showed that nudging the soil moisture (SM) towards the reconstructed fields has

a positive impact on the streamflow May initialisation (Figure 3i) and, therefore, on the associated JJA forecast (Figure 7d). Meanwhile, the effect in boreal winter DJF is negative (Figure 4i and 7h). This effect is consistent with the reduction in the anomaly correlation coefficient (ACC) for both precipitation and near-surface temperature of Figures 5 and 6. The atmospheric impact in summer is lower, as runoff production tends to be driven more by the initial water storage in the basin at the time of forecast initialisation rather than by atmospheric conditions.

The dependence of the atmospheric forecast performance on different land initialisations reveals the critical role of land-atmosphere coupling, which can positively or negatively impact the atmospheric forecast skill. We believe that the negative impact of ICLnud initialisation on the precipitation and temperature seasonal forecasts is linked to the SM nudging, which is expected to improve the variability of soil water content (hence the positive impact on the forecast of river streamflows) but can induce adverse effects on land-atmosphere coupling simulated by the model. For example, the initial soil moisture conditions brought by the offline nudging technique may shift the coupled system away from its equilibrium state. When the forecast integration starts, the nudging constraint is switched off, and the model adjusts to its equilibrium, producing potentially spurious heat and water fluxes (including for the snowpack, if any) at the land-atmosphere interface. Ultimately, this could alter the atmospheric circulation and degrade the temperature and precipitation forecast skill. This makes it difficult to provide further explanations for the regional reductions in ACC noted by the referee (for example, in the northeast of South America or Australia) without performing a deeper dedicated analysis to reveal any causation effect (e.g., Runge et al., 2019).

Hence, Following your suggestion, we propose highlighting the main concerned regions through cyan and red boxes in Figures while modifying the paragraph of the figure discussion to include an explanation of the Online_ICL$_{\mathrm{nud}}$ degradation, as follows.

A global view does not reveal marked changes in terms of ACC for the atmospheric predictions. However, from a continental to regional view, differences are noticeable. In boreal summer (Figure 5), enhanced initialisation ICL$_{\mathrm{nud}}$ tends to increase precipitation correlation in the middle region of South America, including the Paraná River basin and southern Amazon basin (red box), with degradation in the northeast of Brazil, Australia, and some areas of North America and Asia on the north of $40°N$ (cyan boxes). Notably, Europe experiences improved precipitation predictions. Temperature predictions are less sensitive to the land initialisation in summer, but degradation is concentrated in higher latitudes (north of $40°N$ and south of $20°S$). In winter, regions with reduced performance for both precipitation and temperature predictions are primarily found in North Africa, Europe, and Asia (Figure 6).

We have found that the ICL$_{\mathrm{nud}}$ initialisation can harm the accuracy of precipitation and temperature seasonal forecasts in some regions of the globe. This is due to soil moisture nudging, a technique intended to improve the variability of soil water content and the forecast of river streamflows. However, it can also lead to adverse effects on the land-atmosphere coupling simulated by the model. The initial soil moisture conditions introduced by the offline nudging technique may shift the coupled system away from its equilibrium state. When the forecast integration begins, the nudging constraint is deactivated, and the model adjusts to its equilibrium, potentially generating spurious heat and water fluxes at the land-atmosphere interface. This could ultimately alter the simulated atmospheric circulation and reduce the accuracy of the temperature and precipitation forecasts.

[Figure]

**Figure 5.** Comparison of Online_ICL and Online_ICL$_{nud}$ atmospheric forecasts for the anomalies correlation coefficient of the JJA 3-month mean precipitation (a and b) and temperature (c and d). Red (Cyan) boxes highlight regions with noticeable ACC increase (decrease).

[Figure]

**Figure 6.** Comparison of Online_ICL and Online_ICL$_{nud}$ atmospheric forecasts for the anomalies correlation coefficient of the DJF 3-month mean precipitation (a and b) and temperature (c and d). Red (Cyan) boxes highlight regions with noticeable ACC increase (decrease).

4. Line 50: "Conversely, in regions dominated by rainfall, FCAs tend to significantly influence...", what does FCAs mean? This typo has been corrected in the revised version of the manuscript. It is FSCs instead of the typo FCAs.

5. Lines 147-149: "For example, to apply the L3OCV selection method to the hindcast of 1993, only forcing from 1996 to 2020 ensures 25 members. For the hindcast of 2000, only forcing from 1992 to 1998 and 2003 to 2020 is used." The previous article refers to the period from 1993 to 2017, please confirm the range. We appreciate the feedback about the offline configuration reproducing the ESP classical approach. We take the opportunity to correct a typo on the range and clarify the difference between the atmospheric sampling period and the simulation hindcast period. To comply with the atmospheric ensemble in the leave-three-years-out cross-validation framework, we have intentionally avoided using ERA5 atmospheric data from the hindcast year, the previous year and the two subsequent years. As

a result, we have extended the sampling period to include data from 1991 to 2019, which allows us to maintain a sample of 25 atmospheric members to force the offline configuration. Here, the conformation of the atmospheric ensemble follows the ESP approach applied in benchmarking works such as Harrigan et al. (2018) (see reply to comment 2 to see the related changes in the manuscript).

6. Line 172: The values written in the article is not the same as in Figure 2. This typo of the correlation value in the text has been corrected to fit the value in Figure 2. Before (wrong): 0.9986, now (correct): 0.9886.

7. I suggest the colour bar is divided by 0, which makes it possible to visualize the changes in the indicator more clearly in Figs.3-4, and whether the horizontal coordinates of the last column of Figs.3-4 are displayed incorrectly. We understand your suggestion. However, in preliminary versions of figures 3-4 (see second column), we found that dividing the colour bar exactly at zero made it challenging to identify the more significant changes in the skill. Concerning the third column axis, we corrected the horizontal labels in the revised manuscript's last column of Figures 3-4 (see the following reproduction of the reviewed figure for May initialisation).

[Figure]

**Figure 3.** Comparison between May streamflow mean of initialisation run against the observed one over 1993-2017. Left column: ICL bias (a), root mean square error (mm/d) (d), and anomaly correlation (g). Middle column: difference with the $ICL_{nud}$ enhanced land initialisation bias (b), root mean square error (mm/d) (e), and anomaly correlation (h). Right column: distribution of bias for each experiment (c), accumulated distributions of the root mean square (f), and anomaly correlation (i).

**References**

Dunstone, N., Smith, D., Scaife, A., Hermanson, L., Eade, R., Robinson, N., Andrews, M., and Knight, J.: Skilful predictions of the winter North Atlantic Oscillation one year ahead, Nature Geoscience, 9, 809–814, https://doi.org/10.1038/ngeo2824, 2016.

Harrigan, S., Prudhomme, C., Parry, S., Smith, K., and Tanguy, M.: Benchmarking ensemble streamflow prediction skill in the UK, Hydrology and Earth System Sciences, 22, 2023–2039, https://doi.org/10.5194/hess-22-2023-2018, 2018.

Runge, J., Bathiany, S., Bollt, E., Camps-Valls, G., Coumou, D., Deyle, E., Glymour, C., Kretschmer, M., Mahecha, M. D., Muñoz-Marí, J., et al.: Inferring causation from time series in Earth system sciences, Nature communications, 10, 2553, 2019.

**Response to the referee #2 of the research article egusphere-2024-2962: Skilful Seasonal Streamflow Forecasting Using a Fully Coupled Global Climate Model**

Gabriel Fernando Narváez-Campo & Constantin Ardilouze

April 23, 2025

We thank the referee for the detailed review and feedback. We will prepare a revised manuscript addressing the comments. We have organized this reply document as follows:

- The referee comments are in black.

- Our responses are in blue.

- The paraphrases with additions/modifications proposed in the manuscript are in red.

- Figures prepared for the reply have the prefix "R", such as Figure R1.

**Revisions**

1. Section 2.4: More explanation is needed here. Is streamflow bias correction applied only to the online models? If so, how is the comparison fair when offline models are not post-processed?

   We thank you for the opportunity to clarify this key point. The atmospheric quantities are not bias-corrected for any forecasting configuration. Meanwhile, during post-processing, the streamflow bias correction is applied to offline and online forecasts for consistent comparison. Accordingly, this point will be clarified in section 2.4 of the revised manuscript, as follows.

   Our study uses an online atmosphere-ocean-land-river coupled model, for which bias correcting the atmospheric forcing is irrelevant. Instead, we correct the streamflow forecast bias for each flow-gauge station using the Empirical Quantile Mapping method (EQM). To ensure consistent comparisons, we apply streamflow bias correction to both offline and online forecasts.

2. Section 3.1, Figure 3: Why do panels (c, f, i) all use percent mean bias (%) as the x-axis? This makes the figure difficult to interpret. The same issue applies to Figure 4.? Panels (c, f, i) correspond to percent mean bias (%), RMSE (mm/day) and ACC, respectively. We have corrected the horizontal labels in the revised manuscript's last column of Figures 3-4 (see the following reproduction of the reviewed figure for May initialisation).

[Figure]

**Figure 3.** Comparison between May streamflow mean of initialisation run against the observed one over 1993-2017. Left column: ICL bias (a), root mean square error (mm/d) (d), and anomaly correlation (g). Middle column: difference with the $ICL_{nud}$ enhanced land initialisation bias (b), root mean square error (mm/d) (e), and anomaly correlation (h). Right column: distribution of bias for each experiment (c), accumulated distributions of the root mean square (f), and anomaly correlation (i).

3. Line 212: Clarify what is meant by "dryest regions" here. The same applies to Figure 4. We agree that the "dry region" concept was unclear in the referred statement. Based on the global aridity index (see Figure R1), we can better visualize our considerations about dry regions.

[Figure]

**Figure R1.** Global Aridity index comparison against May and November streamflow mean bias of initialisation runs. Left column: Aridity Index (Zomer et al., 2022). Middle column: ICL bias of May. Right column: ICL bias of November.

We propose the following modified paragraph to the original manuscript.

For May, the streamflow Bias of ICL tends to be positive in the dryest regions (Fig. 3a), particularly in western North America, southwestern South America, northeastern Brazil, southern Africa, Iberian peninsula and Australia.

This claim is less evident for November (Fig. 4a). Thus, we have removed the short sentence referring to dry regions without generating any disagreement or contradiction in the discussion, as follows.

The performance of the river initialisation in November (used for DJF forecasts) is presented in Fig. 4. $ICL_{nud}$ tends to reduce the mean bias of stations displaying a high positive bias in ICL  (Figure 4a-b).

4. Line 213: The argument for this figure is not very clear to me. The current phrasing implies that the positive/negative bias directions remain unchanged from ICL to ICLnud, which is not necessarily the case. The reduction of negative bias refers to that some original blue-marked points in (a) got red points in (b), but as I understand, the figure (b) shows the difference in the absolute value of bias, meaning the ICLnud bias can be either negative or positive. While the claim that bias is reduced is still valid. Maybe try to rephrase the argument, and it would be useful to also show the bias of ICLnud, perhaps in an appendix. The bias frequency distribution in Figure 3c further supports our claim. We observed that the peak on the negative bias side (represented by the red curve) shifts toward a smaller bias for the $ICL_{nud}$ (black curve), indicating a decrease in negative bias. On the positive bias side, the black curve rises above the red curve, suggesting an increase in positive bias compared to the ICL. However, this change is observed in only a small percentage of stations, resulting in a minimal overall impact. To clarify this statement, we have rephrased it in the revised manuscript as follows:

The higher concentration of red markers in Figure 3b indicates a reduction in bias from ICL to $ICL_{nud}$. This reduction is more pronounced for negative bias, as displayed by the shift of the negative peak towards zero bias in the frequency distribution shown in Figure 3c.

5. Lines 232-235: The description for locations is inconsistent, sometimes referring to latitude, sometimes to country names, which is difficult to follow. Also, the interpretations themselves sometime do not match with each other. For example, in Line 234, it is stated that Europe shows improved precipitation predictions, but the previous sentence states degradation in the north of 40°N. Similarly, Australia is within the range that is described as showing improved correlation as it is south of 20°S, but it shows degradation in the figure. In general, this figure is difficult to interpret. Consider either adding highlighted boxes on the plot to clearly mark the areas being discussed, or use a better way to specify the area in the text.

We understand the difficulty caused by the inconsistency in the location description. Following the suggestions, we have included boxes indicating the discussed areas in the figures. Besides, the location description has been homogenised, as follows.

A global view does not reveal marked changes in terms of ACC for the atmospheric predictions. However, from a continental to regional view, differences are noticeable. In boreal summer (Figure 5), enhanced initialisation $ICL_{nud}$ tends to increase precipitation correlation in the middle region of South America, including the Paraná River basin and southern Amazon basin (red box), with degradation in the northeast of Brazil, Australia, and some areas of North America and Asia on the north of $40°N$ (cyan boxes). Notably, Europe experiences improved precipitation predictions. Temperature predictions are less sensitive to the land initialisation in summer, but degradation is concentrated in higher latitudes (north of $40°N$ and south of $20°S$). In winter, regions with reduced performance for both precipitation and temperature predictions are primarily found in North Africa, Europe, and Asia (Figure 6).

[Figure]

**Figure 5.** Comparison of Online_ICL and Online_ICL$_{nud}$ atmospheric forecasts for the anomalies correlation coefficient of the JJA 3-month mean precipitation (a and b) and temperature (c and d). Red (Cyan) boxes highlight regions with noticeable ACC increase (decrease).

[Figure]

**Figure 6.** Comparison of Online_ICL and Online_ICL$_{nud}$ atmospheric forecasts for the anomalies correlation coefficient of the DJF 3-month mean precipitation (a and b) and temperature (c and d). Red (Cyan) boxes highlight regions with noticeable ACC increase (decrease).

6. Line 253: "From Online_ICLnud to Online_ICL, the ACC is slightly reduced, especially for basin outlets north of 40°N." Is this correct? It seems like the opposite may be true, please verify. This typo has been corrected, as follows.

    In addition, from Online_ICL to Online_ICL$_{nud}$, the ACC is slightly reduced, especially for basin outlets in the north of $40°N$.

7. Line 255, Figures 7: Are the online system results in this figure bias-corrected or not? Some explanation would help to understand. As stated in Section 2.4 (see reply to comment 1 of this revision) and in the caption of Figure 7, all streamflow forecasts are bias-corrected.

8. Figure 7: The red color is used to represent better performance, which is somehow difficult to remember. Consider either adjusting the color scheme or adding a note in the legend to show the optimal side.

The optimal value and explanation of all the scores computed and evaluated in our study are contained in Table 3 of the manuscript. However, following your suggestion, we have reinforced the optimal side of the anomaly correlation coefficient presented in Figure 7.

[Figure]

**Figure 7.** Anomaly correlation coefficients (ACCs) of bias-corrected streamflow hindcasts computed against observations in JJA (first column) and DJF (second column). Offline_ICL benchmark (First row) and the online coupled configurations with conventional initialisation (second row) and improved initialisation (third row). Cumulative distribution of the anomaly correlation coefficient of the corresponding season (last row). Markers with transparency represent stations with a statistically non-significant ACC at the 95% confidence level.

9. Figure 12: There are red curve lines overlapping with the legend text. This visualization issue will be corrected in the revised manuscript.

10. Line 300:"Arid regions" here, which specific areas are being referred to? This is not clear to me. It was

a typo. We referred to all of North America, not only the arid regions. The new corrected sentence in the manuscript is:

During summer, in North America  (Figure 12a-b), the enhanced initialisation provides about 25% of additional skill (Figure 12c).

11. Line 313: The phrase "remains limited for most conrespecttinents" likely contains a typo. Please check or clarify. This typo has been corrected in the revised version of the manuscript. The right statement is:
    In JJA and DJF, the reduction of RMSE, if any, remains limited for most continents.

**References**

Zomer, R. J., Xu, J., and Trabucco, A.: Version 3 of the global aridity index and potential evapotranspiration database, Scientific Data, 9, 409, https://doi.org/https://doi.org/10.1038/s41597-022-01493-1, 2022.

---

## Editor Decision (ED1)

[revised manuscript text omitted]

**2.3 Streamflow observational database**

Most previous works evaluate the "potential" streamflow predictability of a forecast system by adopting the perfect-model assumption, in which the streamflow forecast is compared to simulated streamflow (from a model driven by meteorological observations) instead of observed streamflow. Meanwhile, we compare the forecasts against observations because, in addition to the IHC and FSC, it incorporates the uncertainty associated with model error (due to structure, physics, and parameter uncertainty) and provides actual (as opposed to potential) streamflow predictability, which is more valuable for end-users or the development of climate services. A database of 1755 flow gauge stations has been created, compiling the global streamflow open access datasets presented in Table 2. We have filtered the full dataset to remove stations with relatively small drainage areas poorly represented by the model resolution and those stations with more than 25% of missing streamflow records in the concerned season.

**Table 2.** Streamflow observed datasets.

[revised manuscript text omitted]

A global view does not reveal marked changes in terms of ACC for the atmospheric predictions. However, from a continental to regional view, differences are noticeable. In boreal summer (Figure 5), enhanced initialisation $ICL_{nud}$ tends to increase precipitation correlation in the middle region of South America, including the Paraná River basin and southern Amazon basin (red box), with degradation in the northeast of Brazil, Australia, and some areas of North America and Asia on the north of $40°N$ (cyan boxes). Notably, Europe experiences improved precipitation predictions. Temperature predictions are less sensitive to the land initialisation in summer, but degradation is concentrated in higher latitudes (north of $40°N$ and south of $20°S$). In winter, regions with reduced performance for both precipitation and temperature predictions are primarily found in North Africa, Europe, and Asia (Figure 6).

We have found that the $ICL_{nud}$ initialisation can have a detrimental effect on the accuracy of precipitation and temperature seasonal forecasts. This is due to soil moisture nudging, a technique intended to enhance the variability of soil water content and improve the forecast of river streamflows. However, it can also lead to adverse effects on the land-atmosphere coupling simulated by the model. The initial soil moisture conditions introduced by the offline nudging technique may shift the coupled system away from its equilibrium state. When the forecast integration begins, the nudging constraint is deactivated, and the model adjusts to its equilibrium, potentially generating misleading heat and water fluxes at the land-atmosphere interface. This could ultimately disrupt the atmospheric circulation and reduce the accuracy of the temperature and precipitation forecasts.

We have shown evidence of the impact of land IHC on the performance of seasonal atmospheric forecasts as proof of the importance of land-atmosphere feedback. In the following section, we will explore the sensitivity of the SSF to enhanced IHCs in a fully coupled global forecast system.

[Figure]

**Figure 5.** Comparison of Online_ICL and Online_ICL_nud atmospheric forecasts for the anomalies correlation coefficient of the JJA 3-month mean precipitation (a and b) and temperature (c and d). Red (Cyan) boxes highlight regions with noticeable ACC increase (decrease).

[Figure]

**Figure 6.** Comparison of Online_ICL and Online_ICL_nud atmospheric forecasts for the anomalies correlation coefficient of the DJF 3-month mean precipitation (a and b) and temperature (c and d). Red (Cyan) boxes highlight regions with noticeable ACC increase (decrease).

**3.3 Impact of initialisation and coupling on streamflow forecast skill**

The hindcast performance of the ESP benchmark (Offline_ICL) is compared against the hindcasts of the fully coupled configurations with two different land initialisations (Online_ICL and Online_ICL$_{nud}$) to determine the contributions of initialisation and land-atmosphere coupling. Unlike online configurations, where the model forecasts the atmosphere, in Offline_ICL, the atmosphere forcing is based on climatology without land-atmosphere feedback. More details on the three configurations can be found in Section 2.2.2.

**3.3.1 Global view**

In summer, the spatial distribution of the anomaly correlation coefficient of the hindcasts compared to the benchmark Offline_ICL reveals a limited effect of the coupling Online_ICL (Figures 7a-b). However, a substantial improvement of the JJA streamflow forecast is achieved with the enhanced initialisation Online_ICL$_{nud}$ (Figures 7c), also drawn by the cumulative distribution in Fig. 7d.

For winter, in the second column of Fig. 7, the coupled hindcasts with both land initialisations yield a remarkable increase of stations with intermediate and high correlation. The cumulative distribution of the ACC, in Fig. 7d, confirms that the number of stations with an ACC greater than 0.5 (0.7) increases to more than 25%(7%). In addition, from Online_ICL to Online_ICL$_{nud}$, the ACC is slightly reduced, especially for basin outlets in the north of $40°N$. It suggests that soil moisture treatment in ICL$_{nud}$ tends to reduce the ability of the system to predict winter streamflow dynamics in basins with strong ice influence. It should be pointed out that a monthly analysis of the performance at different lead times, presented in Figure S4, shows the same conclusions as a 3-month mean analysis of Figure 7d-h.

[revised manuscript text omitted]

---

## Author Response (AR2)

[revised manuscript text omitted]
 1993, forcing of years 1991 and 1996-2019 ensures 25 members. For the hindcast of 2000, forcing from 1991 to 1998 and 2003 to 2019 is used.

155

Before computing the forecast performance scores, the daily streamflow is averaged on a 3-month basis to represent the seasonal mean. The 3-month streamflow mean (JJA and DJF) is assessed across a global dataset of gauged basins with the observational streamflow data described in Section 2.3. To localise the gauging stations in the correct grid pixel of the model river network, we applied  an in-house methodology based on a distance and drainage area station-to-pixel comparison (see Munier and Decharme (2022) for more details and applications).

**Table 1.** Experiments configurations for land initial condition and hindcast production (see Fig. 1).

| | Simulation | Initial condition | | | | Evolution | | | |
|---|---|---|---|---|---|---|---|---|---|
| **ID** | **Description** | **Atm.** | **Ocean** | **Land** | **River** | **Atm.** | **Ocean** | **Land** | **River** |
| | | *Soil moisture reconstruction (SMR)* | | | | | | | |
| SMR | Offline land simulation to reconstruct soil moisture | Disabled | Disabled | Spin-up | Spin-up | Prescribed (ERA5) | Disabled | Free | Free |
| | | *Historical initialisation runs* | | | | | | | |
| ICL | Online coupling with Atm./Ocean nudged to reanalysis | ERA5 | Glorys | Spin-up | Spin-up | Nudged (ERA5) | Nudged (Glorys) | Free | Free |
| $ICL_{nud}$ | ICL nudged to own soil moisture reconstruction SMR | ERA5 | Glorys | Spin-up | Spin-up | Nudged (ERA5) | Nudged (Glorys) | Nudged (SMR) | Free |
| | | *Hindcasts* | | | | | | | |
| Offline_ICL | ESP Benchmark: offline with land initialisation from ICL | Disabled | Disabled | ICL | ICL | Prescribed* (ERA5) | Disabled | Free | Free |
| Online_ICL | Online with land initialisation from ICL | ERA5 | Glorys | ICL | ICL | Free | Free | Free | Free |
| $Online\_ICL_{nud}$ | Online with land initialisation from $ICL_{nud}$ | ERA5 | Glorys | $ICL_{nud}$ | $ICL_{nud}$ | Free | Free | Free | Free |

*The atmospheric ensemble forcing for the Offline_ICL hindcast is constructed from past climate years selected by a leave-three-years-out cross-validation procedure.

160

[Figure]

**Figure 1.** Schematic of offline and online forecast system configurations and corresponding land-river initialisations. ICL: initial condition from the historical run with the online system; $ICL_{nud}$: initial conditions from a historical run with soil moisture  nudged to fields reconstructed from the offline land simulation SMR. As illustrated by the grey-filled arrows, the design of the experiment allows the evaluation of the coupling effect, the initialisation effect or both.

**2.3 Streamflow observational database**

Most previous works evaluate the "potential" streamflow predictability of a forecast system by adopting the perfect-model assumption, in which the streamflow forecast is compared to simulated streamflow (from a model driven by meteorological observations) instead of observed streamflow. Here, we compare the forecasts against observations because, in addition to the IHC and FSC, it incorporates the uncertainty associated with model error (due to structure, physics, and parameter uncertainty) and provides actual (as opposed to potential) streamflow predictability, which is more valuable for end-users or the development of climate services. A database of 1755 flow gauge stations has been created, compiling the global streamflow  open-access datasets presented in Table 2. We have filtered the full dataset to remove stations with relatively small drainage areas poorly represented by the model resolution and those stations with more than 25% of missing streamflow records in the concerned season.

**Table 2.** Streamflow observed datasets.

| Dataset[a] | Region | Reference |
|---|---|---|
| GRDC: Global Runoff Data Centre | Global | http://www.bafg.de/GRDC/EN/Home/homepage_node.html |
| USGS: United States Geological Survey | United States | http://waterdata.usgs.gov/nwis/sw |
| HYDAT: National Water Data Archive | Canada | https://collaboration.cmc.ec.gc.ca/cmc/hydrometrics/www/ |
| French Hydro database | France | http://www.eaufrance.fr |
| Spanish Hydro database | Spain | http://ceh-flumen64.cedex.es/anuarioaforos/default.asp |
| HidroWeb | Brazil | http://www.snirh.gov.br/hidroweb/ |
| R-ArcticNet | Northern High Latitudes | http://www.r-arcticnet.sr.unh.edu/v4.0/AllData/index.html |
| Australian Bureau of Meteorology | Australia | http://www.bom.gov.au/metadata/19115/ANZCW0503900339 |
| China Hydrology Data Project | China | Henck et al. (2011) |
| HyBAm | Amazon basin | https://hybam.obs-mip.fr/ |

[a]In case of overlapping stations with the global GRDC dataset, priority is given to the national database.

We conducted a correlation analysis to select the minimum drainage area considered in the study. For basins with an area higher than a certain threshold ($A_{threshold}$), Figure 2 shows the correlation between the basins area ($A_{basin}$) and the area estimated for the CTRIP routing model ($A_{CTRIP}$). With increasing $A_{threshold}$, the correlation increases, but the number of available basins reduces. The threshold is set to $6 \times 10^3$ km$^2$ (about two CTRIP cells per basin in mid-latitudes) to maintain a balance between the number of basins analysed and their geometrical representation and to avoid considering basins

inside the spurious oscillating correlation curve (Figure 2). There are 1451 gauged basins with $A_{basin} \geq A_{threshold}$ with a $A_{basin}|A_{CTRIP}$ correlation of 0.9886.

[Figure]

**Figure 2.** Correlation coefficient of $A_{real}|A_{CTRIP}$ and number of basins with area higher than a given $A_{threshold}$.

From the 1451 streamflow stations, we only consider those with less missing data than $25\%$ of the total data in the analysed season. Figure S1 in the Supplement shows the distribution, in  space and frequency, of the full database and the selected stations. The final dataset has 1071 stations in JJA and 1043 stations in DFJ, distributed in North America ($\approx 82\%$), Europe ($\approx 13\%$), South America ($3.5\%$), Africa ($1.7\%$), Asia and Australia ($0.4\% = 4$ stations). In Section 3.3, we remove 14 stations where  the mean bias magnitude exceeded the maximum machine number in double precision. This can occur in basins outlets near the ocean, where backflow can generate close to zero mean observed streamflow (on the denominator of the mean bias formula - see Table 3).

**2.4 Streamflow bias correction**

Typically, statistical post-processing methods are applied to compensate for errors in model structure or initial conditions, correct biases, and improve ensemble dispersion (Troin et al., 2021). Such bias correction can be applied to atmospheric forecasts (such as precipitation, temperature, and evaporation) and/or to hydrological forecasts like runoff and streamflow (e.g., Petry et al., 2023; Tiwari et al., 2022; Gubler et al., 2020; Crochemore et al., 2016; Wood and Schaake, 2008). Our study uses an online atmosphere-ocean-land-river coupled model, for which bias correcting the atmospheric forcing is irrelevant. Instead, we correct the streamflow forecast bias for each flow-gauge station using the Empirical Quantile Mapping method (EQM). To ensure consistent comparisons, we apply streamflow bias correction to both offline and online forecasts.

[revised manuscript text omitted]
 driest regions (Fig. 3a), particularly in the west of North America, Northeastern Brazil, south of Africa, Iberian peninsula and Australia. The higher concentration of red markers in Figure 3b suggests a reduction in bias from ICL to $ICL_{nud}$. This reduction is more pronounced for negative bias, as indicated by the shift of the negative peak towards zero bias in the frequency distribution shown in Figure 3c. Besides, the RMSE is generally smaller with $ICL_{nud}$, in particular over regions with large RMSEs in ICL (Figure 3d-f). In seasonal forecasts, the temporal correlation between the forecasted and the observed anomalies is crucial since it indicates the capability of capturing the inter-annual variability of streamflow departures from the mean value. The spatial distribution of the difference in anomaly correlation coefficient $|ACC_{ICL} - 1| - |ACC_{ICL_{nud}} - 1|$ in Fig. 3h shows that the soil moisture nudging improves the temporal dynamics of the simulated streamflow in May over most of the 1067 gauging stations. The result is verified in Fig. 3i, which reports up to 20% more stations with $ACC > (0.4 - 0.6)$.

The performance of the river initialisation in November (used for DJF forecasts) is presented in  Figure S3. $ICL_{nud}$ tends to reduce the mean bias of stations displaying a high positive bias in ICL (Figure S3a-b). The  global distribution of bias in Fig.  S3-c confirms a reduction of high positive bias, favouring the concentration of bias values closer to zero than ICL. However, unlike JJA, in DJF, $ICL_{nud}$enhances the number of basins with higher

RMSE and lower ACC. In Section 3.3, we show and discuss the impact of the initial hydrologic condition (IHC) degradation on the hindcasts in boreal winter.

[Figure]

**Figure 3.** Comparison between May streamflow mean of initialisation run against the observed streamflow over 1993-2017. Left column: ICL bias (a), root mean square error (mm/d) (d), and anomaly correlation (g). Middle column: difference with the ICL$_{nud}$ enhanced land initialisation bias (b), root mean square error (mm/d) (e), and anomaly correlation (h). Right column: distribution of bias for each experiment (c), accumulated distributions of the root mean square (f), and anomaly correlation (i).

**3.2 Precipitation and temperature skill**

One way to bring out the influence of the land-atmosphere coupling is to assess the impact of different land IHCs on the
atmospheric forecast. The performance of the atmospheric seasonal forecast is presented in Figures 4 and 5, in particular, for
two of the most important water cycle drivers: precipitation and near-surface temperature. Precipitation is compared against
the Multi-Source Weighted-Ensemble Precipitation (MSWEP v2, Beck et al. (2019)) and the temperature against the Climatic
Research Unit gridded Time Series (CRU TS v4.05, Harris et al. (2020)).

A global view does not reveal marked changes in terms of ACC for the atmospheric predictions. However, from a continental
to regional view, differences are noticeable. In boreal summer (Figure 4), enhanced initialisation $ICL_{nud}$ tends to increase
precipitation correlation in the middle region of South America, including the Paraná River basin and southern Amazon basin
(red box), with degradation in the northeast of Brazil, Australia, and some areas of North America and Asia  north of
$40°N$ (cyan boxes). Notably, Europe experiences improved precipitation predictions. Temperature predictions are less sensitive
to the land initialisation in summer, but degradation is concentrated in higher latitudes (north of $40°N$ and south of $20°S$).
In winter, regions with reduced performance for both precipitation and temperature predictions are primarily found in North
Africa, Europe, and Asia (Figure 5).

We have found that the $ICL_{nud}$ initialisation can have a detrimental effect on the accuracy of precipitation and temperature
seasonal forecasts. This is due to soil moisture nudging, a technique intended to enhance the variability of soil water content
and improve the forecast of river streamflows. However, it can also lead to adverse effects on the land-atmosphere coupling
simulated by the model. The initial soil moisture conditions introduced by the offline nudging technique may shift the coupled
system away from its equilibrium state. When the forecast integration begins, the nudging constraint is deactivated, and the
model adjusts to its equilibrium, potentially generating misleading heat and water fluxes at the land-atmosphere interface. This
could ultimately disrupt the atmospheric circulation and reduce the accuracy of the temperature and precipitation forecasts.

We have shown evidence of the impact of land IHC on the performance of seasonal atmospheric forecasts as proof of the
importance of land-atmosphere feedback. In the following section, we will explore the sensitivity of the SSF to enhanced IHCs
in a fully coupled global forecast system.

**3.3 Impact of initialisation and coupling on streamflow forecast skill**

The hindcast performance of the ESP benchmark (Offline_ICL) is compared against the hindcasts of the fully coupled config-
urations with two different land initialisations (Online_ICL and Online_$ICL_{nud}$) to determine the contributions of initialisation
and land-atmosphere coupling. Unlike online configurations, where the model forecasts the atmosphere, in Offline_ICL, the
atmosphere forcing is based on climatology without land-atmosphere feedback. More details on the three configurations can
be found in Section 2.2.2.

[Figure]

**Figure 4.** Comparison of Online_ICL and Online_ICL$_{nud}$ atmospheric forecasts for the  anomaly correlation coefficient of the JJA 3-month mean precipitation (a and b) and temperature (c and d). Red (Cyan) boxes highlight regions with noticeable ACC increase (decrease).

[Figure]

**Figure 5.** Comparison of Online_ICL and Online_ICL$_{nud}$ atmospheric forecasts for the anomalies correlation coefficient of the DJF 3-month mean precipitation (a and b) and temperature (c and d). Red (Cyan) boxes highlight regions with noticeable ACC increase (decrease).

**3.3.1 Global view**

In  summer, the spatial distribution of the anomaly correlation coefficient of the hindcasts compared to the benchmark
Offline_ICL reveals a limited effect of the coupling Online_ICL (Figures 6a-b). However, a substantial improvement of the
JJA streamflow forecast is achieved with the enhanced initialisation Online_ICL$_{nud}$ (Figures 6c), also drawn by the cumulative
distribution in Fig. 6d.

For winter, in the second column of Fig. 6, the coupled hindcasts with both land initialisations yield a remarkable increase of
stations with intermediate and high correlation. The cumulative distribution of the ACC, in Fig. 6d, confirms that the number of
stations with an ACC greater than 0.5 (0.7) increases to more than 25%(7%). In addition, from Online_ICL to Online_ICL$_{nud}$,
the ACC is slightly reduced, especially for basin outlets  to the north of $40°N$.  This suggests that soil moisture
nudging in ICL$_{nud}$ tends to reduce the ability of the system to predict winter streamflow dynamics in basins with strong ice
influence. It should be pointed out that a monthly analysis of the performance at different lead times, presented in Figure S5,
shows the same conclusions as a 3-month mean analysis of Figure 6d-h.

[revised manuscript text omitted]

The greatest improvement in South American rivers for both seasons comes from the dynamic atmospheric forecast incorporated in the coupled systems. Due to the few gauge stations in Africa, the cumulative distribution does not provide robust information. As a result, the different levels of coupling and initialisation do not show evidence of impact on the seasonal prediction of streamflow in the 15-17 gauging stations evaluated in Africa.  Besides, most of the (few) gauges in Africa are in the Southern part of the continent, where JJA is the dry season, while DJF is positioned in the (monsoonal) wet season. Under this data availability context for Africa, 
[revised manuscript text omitted]
 potential predictability of the system using the perfect model approach. In this framework, the river streamflow historical time series derived from the initialisation run is used as a verification dataset instead of actual observations. This approach allows first to remove the influence on river discharges of human activity that is not parameterized in the model (dams, reservoirs, irrigation). Secondly, the forecast evaluation can be performed on every location where the model simulates a streamflow. This allows to define a set of virtual stations equally distributed across the globe and thereby assess the predictions also in regions poorly instrumented.

Our study also provides insights for improving the forthcoming generation of forecast systems for hydrological predictions, particularly regarding initialisation methods.

In light of our results, we aim to explore other seasons and develop a novel and more robust land-river initialisation with more realistic soil moisture conditions. This strategy is currently being tested *via* the in-house land data assimilation system LDAS-Monde (Albergel et al., 2020) in the context of the Horizon Europe project CERISE (https://www.cerise-project.eu/). Finally, in the longer term, we expect forecast improvement from better representation of the influence of human activity on the terrestrial water cycle. In particular, we will evaluate the activation of the novel irrigation scheme (Decharme et al., 2025) within a higher-resolution version of CTRIP (Munier and Decharme, 2022) in the CNRM-CM GCM of next-generation for CMIP7, focusing on river streamflow seasonal predictions.

[revised manuscript text omitted]